# A Review of Multi-Material 3D Printing of Functional Materials via Vat Photopolymerization

**DOI:** 10.3390/polym14122449

**Published:** 2022-06-16

**Authors:** Usman Shaukat, Elisabeth Rossegger, Sandra Schlögl

**Affiliations:** Polymer Competence Center Leoben GmbH, Roseggerstrasse 12, 8700 Leoben, Austria; usman.shaukat@pccl.at (U.S.); elisabeth.rossegger@pccl.at (E.R.)

**Keywords:** 3D printing, vat photopolymerization, multimaterials, step-growth, chain-growth, cationic polymerization, orthogonal networks, grayscale printing

## Abstract

Additive manufacturing or 3D printing of materials is a prominent process technology which involves the fabrication of materials layer-by-layer or point-by-point in a subsequent manner. With recent advancements in additive manufacturing, the technology has excited a great potential for extension of simple designs to complex multi-material geometries. Vat photopolymerization is a subdivision of additive manufacturing which possesses many attractive features, including excellent printing resolution, high dimensional accuracy, low-cost manufacturing, and the ability to spatially control the material properties. However, the technology is currently limited by design strategies, material chemistries, and equipment limitations. This review aims to provide readers with a comprehensive comparison of different additive manufacturing technologies along with detailed knowledge on advances in multi-material vat photopolymerization technologies. Furthermore, we describe popular material chemistries both from the past and more recently, along with future prospects to address the material-related limitations of vat photopolymerization. Examples of the impressive multi-material capabilities inspired by nature which are applicable today in multiple areas of life are briefly presented in the applications section. Finally, we describe our point of view on the future prospects of 3D printed multi-material structures as well as on the way forward towards promising further advancements in vat photopolymerization.

## 1. Introduction

Polymers have revolutionized the world with their diversity, tempting appearances, inherent properties, and unique functionality. They are macromolecules formed by repeating monomer units, and thus exhibit exceptional physical, mechanical, and viscoelastic properties. Because of their long chemically bonded macromolecular chains, mechanical and thermal stimuli can induce mobility and deformations in polymers [1].

The vast variety of monomers and their adjourning mechanisms have led to polymers differing in flexibility, stiffness, and elasticity. Based on their crosslinked structure and related thermo-mechanical and viscoelastic properties they can be divided into thermoplastics, elastomers, and thermosets, which collectively represent a huge share of the technically relevant materials available on the market [2,3,4]. As the name suggests, thermoplastics are high molecular weight polymers; they do not possess chemical crosslinks, and thus, at elevated temperatures, they become mobile, soft, and flowable enough to be shaped into commodity articles [3,5,6]. Cooling the shaped articles below the glass and melting temperature hardens thermoplastic objects and fixes the microstructure. Thermoplastics can be processed by injection molding and extrusion; upon heating, they can be softened, melted, and reshaped repeatedly. In contrast, duromers, thermosets, or thermosetting polymers are highly crosslinked polymer networks, which harden and cure into a permanent shape during processing [7]. Such curing reactions are typically induced by heat or ultraviolet-visible (UV-Vis) light irradiation, and form irreversible crosslinks within the polymer structure [2]. Owing to their highly crosslinked nature and the inherent rigidness of the polymer network the recycling of duromers is challenging, as they are infusible and non-soluble. Elastomers contain chemical crosslinks as well, albeit at a lower degree than duromers. They are soft rubber-like polymers with a very low glass transition temperature and notably lower Young’s modulus. They are typically known for their exceptional stretchability beyond their original dimensions [4,8]. Elastomers are interchangeably denoted as rubbers due to their inherent elasticity and amorphous nature, which induces considerable segmental mobility within the chemical bonds [9].

The mechanical properties, especially the stiffness of the final polymer product, is greatly affected by its composition, macromolecular structure, and number/type of crosslinks. The stiffness of a polymer is a measure of the ratio of the applied force to the strain produced in the material. Technically expressed as the Young’s modulus (*E*), it is defined by the tangent of the modulus in the linear elastic region, and is typically determined at an initial strain < 2% [10].

The Young’s modulus of polymers represents an important property that discriminates between soft and stiff materials. Table 1 compares the Young’s moduli of selected polymers with other material classes. Elastomers and hydrogels are soft, with a Young’s modulus in the range of 10^3^–10^9^ Pa, which is comparable to the rigidity of natural soft tissues such as muscle or skin. Thus, these classes of materials dominate sectors such as biomedical engineering [11,12,13], soft robotics [14,15,16,17,18], and flexible electronics [19,20,21].

In contrast, structural thermoplastic polymers have a Young’s modulus between 10^9^–10^12^ Pa, which is in the range of the Young’s modulus of wood or bone [1,14].

Various processing strategies have been adopted to date for the manufacturing of multi-material structures, including both soft and rigid domains. This particular interest in multi-material manufacturing is inspired by nature, where countless exemplary multi-material architectures of soft and hard constituents, including nacre [23], shell [24], wood [25], bone [26,27], and crustacean exoskeleton [28], exhibit unique features in terms of mechanical properties and functionality.

Consolidation of a wide spectrum of materials could form materials with large variations in properties or material functionalities in order to achieve required features [29,30]. However, this can generate significant interfacial stresses, leading to an interface vulnerable to mechanical failure [31,32]. Significant problems have been recorded in application areas employing hydrated polymers such as hydrogels, as they can undergo swelling between heterogenous materials, generating additional internal stresses [33]. The most common problems encountered as a result of internal stress involve delamination and localized crack failures. Consequently, manufacturing and application of synthetically formed soft and stiffer materials is a prerequisite for upgraded processing methodologies [34].

Additive manufacturing, often termed 3D printing, is a comprehensive materials development technique which has gained signification attention following its discovery in the 1980s [1,2,3,4,5,6,35,36,37]. At present this technology is experiencing significant growth and can now be applied using a wide range of methods, including material jetting [38,39], binder jetting [40,41], material extrusion [42,43,44,45,46], powder bed fusion [47,48], vat photopolymerization [37,49,50,51,52,53], and sheet lamination [35,36,54,55].

In particular, additive manufacturing or 3D printing of polymers offers significant advantages over conventional processing methodologies for the formation of 3D objects with multi-material properties. The single-step manufacturing of complex three-dimensional products offers drastically shortened processing times, and has gained increased attention over the past decade [4,24,35,46,49,56,57,58,59,60,61,62,63,64,65,66,67,68,69,70].

The designed structures are evolved layer-by-layer or point-by-point via a computerized controlled system in order to replicate the desired 3D shape models. Contrary to traditional processing methods, which require the use of dies, molds, or lithography masks, 3D printing technologies employ automated assemblies that translate the computer-aided designs into sophisticated 3D structures without excess loss of material [71,72,73,74].

Furthermore, the ability to quickly develop final products with lower process complexity along with the lower costs of printing technologies has attracted both academia and industry. To date, more than fifty different types of additive manufacturing technologies have been reported on the basis of the processing principles [1]. The choice of the technology being used for the manufacturing of a specific product greatly depends on the physical and chemical properties of the raw materials and the targeted end applications of the final product.

Because thermoplastics become repeatedly soft and processable above their melting temperature, they are processable by extrusion based additive manufacturing technologies [3]. However, due to their limited mechanical properties, they are not always suitable for structural applications in neat condition, and it is often essential to add particular additives such as reinforcing fillers in order to adjust the properties of the 3D printed articles [75,76]. However, due to the absence of covalent bonds, 3D printed thermoplastics can be easily reprocessed and re-melted.

In contrast, 3D printed thermosets and duromers cannot be reprocessed, as chemical crosslinks are formed during the printing process. These curing reactions are generally irreversible; once covalently crosslinked, the materials are insoluble and infusible. Curing reactions are typically triggered by temperature or light. In particular, optically triggered curing reactions have several advantages, such as fast curing under mild conditions (even at room temperature), spatial control of the reaction, and low energy consumption [77,78,79]. These make photoreactions ideal candidates for additive manufacturing. In vat photopolymerization 3D printing, a liquid photocurable resin is selectively cured by light exposure and the structure is formed layer-by-layer (digital light processing 3D printing) or point-by-point (stereolithography). These vat photopolymerization additive manufacturing technologies employ a variety of different functional monomers including epoxy resins, phenolic polymers, unsaturated polyesters, acrylates, methacrylates, and organosilicons [80]. Additive manufacturing of such systems can generate both soft materials and photopolymers with high mechanical strength and stiffness, similar to that of dental resins [81].

Hydrogels present another subclass of polymer networks; their high hydrophilicity, biocompatibility, and great flexibility make them ideal candidates for tissue engineering applications [82,83]. Chemical crosslinking methodologies govern the structural development of hydrogels and maintain the mechanical integrity of hydrogel materials and scaffolds [84,85].

Among the established additive manufacturing technologies, vat photopolymerization technologies outperform in terms of their excellent building resolution, dimensional accuracy, diversified reaction chemistries, low equipment costs, and high surface quality [49,53,86]. Thanks to the attractive features of vat photopolymerization, recent advancements have extended materials fabrication capacity; numerous different techniques have been established, including stereolithography (SLA), digital light processing (DLP), liquid crystal display (LCD) printing, continuous liquid interface printing (CLIP), and two-photon absorption (TPA) printing [37,49,50].

In this review paper, we primarily focus on vat photopolymerization-based 3D printing technologies, discussing in detail their manufacturing processes, accessible feedstock chemicals, and underlying curing mechanisms. Furthermore, we exclusively target the advancements in multi-material vat photopolymerization and its ability to spatially control mechanical properties through a combination of multiple materials and reaction chemistries within 3D printed architectures. In a systematic way, we develop the understanding of the whole multi-material vat photopolymerization domain, then present several recent advances.

A growing number of experts and researchers with diverse backgrounds have begun to utilize additive manufacturing technologies for developing complex objects with multi-material properties which cannot be produced using traditional processing techniques. Many remarkable results have been reported over the last couple of years [87,88], offering significant potential to revolutionize the next generation of functional materials. Furthermore, the number of publications in the last decade has increased significantly (Figure 1), representing a great perception of the emerging possibilities in this research field. At the moment, extrusion-based technologies (fused filament fabrication, direct ink writing) are mostly employed for additive manufacturing of polymers, although they suffer from a lower spatial resolution and poor multi-material building capabilities at the molecular scale (e.g., poor adhesion at the interface between soft and rigid domains) [49]. Here, vat photopolymerization technologies could offer the required spatial and temporal control, allowing nanoscale interactions between the reacting precursors.

Nonetheless, multi-material vat photopolymerization 3D printing remains in the early development phases and faces several challenges, including: (1) limited functionality of monomers; (2) a limited library of photopolymers supporting multi-material printing technology; (3) loss of orthogonality in reactions; (4) leaching of materials and long-term property maintenance; and (5) the viscosity and printability of precursors with a suitable resolution.

In order to highlight the full capabilities of multi-material vat photopolymerization 3D printing, the multi-material printing process itself and the vat photopolymerization strategies are both addressed in this review. At the end, we emphasis the application-oriented aspects of multi-material vat photopolymerization 3D printing in flexible electronics, biomedical applications, soft robotics, and rapid prototyping. In addition, we discuss the latest concepts around increasing the sustainability and recyclability of highly crosslinked 3D structures which rely on the introduction of dynamic covalent bonds, and elaborate the postprocessing, mechanical properties, and economical aspects of the technology.

## 2. Additive Manufacturing Technologies

Additive manufacturing or 3D printing of polymers can be generally classified into extrusion-based technologies (fused filament extrusion, direct ink writing), polymer bed sintering, vat photopolymerization-based technologies (stereolithography, digital light processing, liquid crystal display printing, continuous liquid interface printing, two-photon absorption printing), and 3D-volumetric polymerization technologies. Each technology has its own advantages and limitations, and the choice of a processing methodology depends on the feed materials, required printing resolution, printing times, size, performance of the printed materials, and overall cost of fabrication [89,90,91,92].

In this section, we briefly illustrate the different processing techniques and highlight their key features; an executive summary is provided at the end.

### 2.1. Material Extrusion (Fused Filament Fabrication)

Fused filament fabrication (FFF) is the most widely used 3D-printing technology for thermoplastic polymers. In this process, polymeric filaments are fed into the extrusion head, where they are melted into a semi-liquid state. Acrylonitrile butadiene styrene (ABS), polylactic acid (PLA), and polycarbonate (PC) are the most widely used thermoplastic filaments due to their low processing temperatures. The molten polymers are then extruded through the nozzle on the building platform, where the melt is allowed to cool down and fuse together (Figure 2). The motion of the extrusion nozzle is controlled layer-by-layer according to the 3D design interpretation.

Adjustment of printing parameters such as temperature, building orientation, air gap, raster angle and width, and thickness of layers plays a vital role in the quality of 3D printed articles [10,93]. A major constraint when utilizing FFF technology is the need to feed the polymers in filament form. The requirements of high temperatures, an acceptable viscosity range for extrusion, and homogenous and void-free melt flow represent challenging areas in FFF. On the other hand, the lower cost of printing, simplicity of the process, and fast building speed make FFF a very attractive technology for the additive manufacturing of 3D objects.

### 2.2. Direct Ink Writing

Direct ink writing (DIW) involves the extrusion of viscoelastic liquid pastes through a pressurized nozzle. The material is extruded on a fixed platform and allowed to cool down and solidify, while the nozzle is able to move and defines the shape of the printed material (Figure 3). The final structure of the material is fixed via layer-by-layer deposition of the liquid polymer ink [69]. The viscosity of the extruded liquid and the speed of extrusion defines the quality of printed architectures [94]. The primary benefit of this technology is its material versatility, as polymer solutions, hydrogels, and pastes can be easily charged into the extrusion nozzle. A sacrificial structural support may be required for the fabrication of complex architectures, as the viscoelastic nature of the material can collapse the 3D object during building.

### 2.3. Selective Laser Sintering

In the selective laser sintering (SLS) 3D printing technique, a solid polymer powder bed is locally heated with a scanning laser. Laser illumination causes the melting and fusion of the polymeric powder, which is spread over the surface of the build platform by a rolling assembly. This spatially controlled exposure to laser light causes the melting of the polymer at selected regions. For the next layer, a piston assembly moves one step down to accumulate the polymer powder with the help of a spreading roller (Figure 4). The whole process is repeated until the required number of layers according to the printing program is achieved [95]. With powerful high-energy laser scans, the adjacent polymer particles are able to fuse and molecularly diffuse into the printed structure. The removal of unbounded materials during a postprocessing step is essential to achieving good resolution of the printed articles.

The size of the polymeric powder particles, scanning speed, scan width and gap, and laser intensity all have an impact on the final structure of printed materials [96]. In SLS technology, any thermoplastic powder could ideally be fused and processed; however, the molecular diffusion, complicated integration, and sintering of particles greatly limit the number of processable polymers [97]. Widely used polymers in SLS include polyamide and polycaprolactone.

### 2.4. Stereolithography

Stereolithography (SLA) is a vat photopolymerization process in which liquid resins with adequate viscosity are introduced into a vat. Light is illuminated on the liquid resin in a controlled manner, resulting in polymerization reactions at the targeted positions (Figure 5). The 3D structure is evolved by illumination of the 3D design point-by-point, guided by 3D interpretation software [57,98,99]. Liquid resins such as methacrylates, acrylates, vinyl, and epoxy monomers are typically used in SLA technology. The quality of the printed structures depends on the cure kinetics of the polymerization process, which is guided by the light intensity, illumination time, resin viscosity, chemical functionality, and the additives in the formulations. Photoinitiators can be introduced into formulations in order to initiate the reactions, while light absorbers can improve the resolution of printed objects.

The key benefit of SLA technology is the very high resolution of the printed objects. Furthermore, as the monomers are already in liquid form, heating of the polymer feed and nozzle is not required. Unfortunately, the technology is limited by the number of available photopolymers. Cytotoxicity and the irritation potential of the resins and the unreacted monomers in the printed articles should be considered while 3D printing using SLA [100].

### 2.5. Digital Light Processing

Digital light processing (DLP) is another class of vat photopolymerization technology, in which illumination with light starts the polymerization reaction locally. Liquid resins containing vinyl, acrylate, or epoxide groups are often employed, along with a suitable photoinitiator and a light absorber. In contrast to SLA, the liquid resin is irradiated layer-by-layer according to the design of the printed article (Figure 6). Photoinitiators absorb the light within a certain wavelength range according to their absorption capability, triggering the polymerization reaction in the illuminated regions. As polymerization takes place, the final printed article is formed by the solidification of each layer. After printing, the object is removed from the build platform and the remaining resin is recovered from the vat [101,102].

The key benefits of DLP 3D printing are the high resolution of the final product and the fast building speed. As in SLA, DLP technology is limited by the availability of monomers that can be photopolymerized.

### 2.6. Liquid Crystal Display

Liquid crystal display (LCD) is another a vat photopolymerization technology, which uses a liquid crystal display unit for the imaging tasks. By applying an electric field, molecular rearrangements of the liquid crystals take place, which block the passage of light through selected areas (Figure 7). The design of the 3D printed structure is translated through a computerized system, which creates a response in the LCD imaging unit. Very high printing resolutions can be achieved with the leading LCD technologies. Despite the promise of LCD technology, molecular rearrangements can remain trapped or stuck under the applied electrical field, which can enable the passage of light through the LCD screen, resulting in reduced resolution [103,104,105].

### 2.7. Continuous Liquid Interface Printing

Continuous liquid interface printing (CLIP) technology was first introduced in 2015, and permits a continuous printing of monolithic and layer-less photopolymeric architectures [106]. The technology is based on the utilization of an oxygen-permeable chamber that allows the formation of an interfacial oxygen layer. The radical induced polymerization reaction is inhibited by the presence of the oxygen layer. This oxygen interference is able to quench the excited phase of photoinitiators, and can form peroxides by reacting with free radicals in the polymerizing system [107]. As a result, unreacted monomers remain at the interface between the build platform and oxygen layer. This results in rapid printing of polymer resins without slicing the 3D designs, avoiding the conventional layer-by-layer approach (Figure 8). CLIP technology can typically draw out printed objects at a very fast speed, with production cycles of only a few minutes and a resolution lower than 100 µm [106,108,109]. However, it is limited to curing systems which are quenched by oxygen, such as photocurable (meth)acrylate-based monomers.

### 2.8. Hot Lithography

The hot lithography process employs the principles of vat photopolymerization technology, with the difference that the vat is additionally supplied by a heating element to control the temperature of the printing process (Figure 9). During conventional vat photopolymerization processes the printing temperature is kept at room temperature, which is sometimes limited by the higher viscosity of the printing resins. By taking advantage of higher-temperature processing, hot lithography can reduce the viscosity of the resins while increasing the reactivity of monomers, which cannot be fully polymerized at room temperature [110,111,112]. Photocurable resins are cured by applying a typical SLA or DLP projector, although the conversion of monomers can be expected to be higher in comparison to room temperature curing [111]. This technology offers the potential to enhance the resin portfolio available for vat photopolymerization.

### 2.9. Two-Photon Absorption 3D Printing

Two-photon absorption 3D printing (TPA or TPP) is a vat photopolymerization technique which employs a highly compact laser beam as a light source. An extremely high-powered density of illumination, in the range of 10^13^ W/µm^2^, on a very confined volume of less than a cubic wavelength (λ^3^) results in photon absorption in a nonlinear pattern (i.e., two-photon absorption) [113]. Hence, an excellent spatial resolution (i.e., nanometer range), far ahead of the optical diffraction limit, can be realized [114]. To take complete advantage of non-linear absorption, an infrared irradiation source is adopted to facilitate the deep penetration of laser light into the bulk of the matter, minimizing absorption power loss. In the case of lithography where the light is linearly exposed, the material response only corresponds to the first order. However, the response is of the second or multiple orders as a result of two-photon or multi-photon absorption, respectively (Figure 10). The square light intensity distribution is spatially more precise than the linear one [114], consequently decreasing the light–matter interaction volume and resulting in improved printing resolution. The photoinitiator absorbs two photons in the near-infrared range (NIR), providing a similar quantum of energy to that supplied by the single-photon absorption in a photocurable formulation. Hence, the spatially distributed photoinitiated radicals formed during two-photon polymerization of resins follow the square law of light intensity function. Due to the presence of radical scavengers such as dissolved oxygen in the resin formulations, radicals are sustained in the regions of higher light intensity and a polymerization threshold is attained. This light intensity corresponding to the threshold is adjustable using printing parameters such illumination time, target unit volume (i.e., voxels), and illumination intensity. Using TPA, printing resolutions much lower than the defined diffraction limit can be achieved [115,116,117,118,119,120]. With adjustments in the 3D printing parameters, voxels up to 100 nm have been 3D printed [114,121,122].

TPA 3D printing can be used to fabricate nanostructures that cannot be developed with other technologies. Owing to a very strong transient power, laser illumination in TPA can activate multiple reactive species in the resins; hence, a suitable strategy is required in order to effectively use the process for the 3D printing of nanostructures (Table 2).

### 2.10. Volumetric 3D Printing

Volumetric 3D printing is a process in which multiple sliced images are DLP projected at distinct angles on a synchronized rotating resin chamber according to computer-aided designs [128,129]. While the intensity of a single illuminated ray is deficient enough to avoid the photopolymerization of resin under oxygen inhibition, the superposition of exposed irradiation leads to the activation of radical species above a threshold, resulting in the polymerization of a centimeter-scale object in a matter of few seconds. The degree of polymerization for each resin can be adjusted by tuning the light intensity and the exposure duration. For such volumetric techniques, resins with a higher viscosity (up to 90 Pa.s) can be 3D printed [128]. Typically, flexible polymers with an elastic modulus of <10 kPa and discrete applications such as bio-imprints and tissue engineering are challenging to print using layer-by-layer printing technologies. Here, gravitational and auxiliary forces occur during printing, which result in frequent breakage of the layers and building structure. Volumetric 3D printing can easily overcome these problems by avoiding the layer-by-layer approach. A resolution of 80 µm has been achieved [130] using volumetric printing technologies, with a printing speed much higher than conventional vat photopolymerization-based technologies. There are many classes of volumetric 3D printing techniques, including computer axial lithography (CAL), xolography [86], tomographic printing [129,130], and holographic light patterning [131]. While the technology remains very much under development, a few of these, such as xolographic and tomographic printers (Readily 3D), are already commercially available; however, they have not yet been extended to multi-material 3D printing.

In contrast, solution mask liquid lithography (SMaLL) is a volumetric 3D printing technology which has shown the potential to fabricate 3D objects with multi-material properties. In this technology, collimated UV-Vis light is illuminated over an optically dense resin consisting of a combination of photocurable monomers, photoinitiators, photochromes, and photosensitizers. Particular interest has been generated by the addition of photochromes to deploy photo-bleaching fronts capable of activating the photosensitizers and subsequently deep curing the materials, which significantly enhances the building speed. During 3D printing, the reversible absorption characteristics of the photochromic dyes are exploited to locally activate or deactivate photoreactions at different wavelengths [24].

The detailed summary of all 3D printing techniques is also added in Table 3.

## 3. Multi-Material 3D Printing

Multi-material 3D printing involves the combination of two or more material properties in a single 3D printed article. The combination of multiple materials can be accomplished through different printing technologies. Among the class of extrusion-based 3D printing techniques, fused filament fabrication (FFF) is the most widely used 3D printing technology. The first dual-component FFF was proposed in 2002 via the modeling of an optimization strategy for FF technology [148]. Subsequently, numerous studies have been reported on the development and fabrication of multi-material structures with FFF [60,67,149,150]. A schematic illustration of a multi-material FFF printer is shown in Figure 11. Multiple nozzles are installed under the extrusion head for melting and ejecting each polymer filament onto the build platform. After a desired number of layers is printed, the extrusion process is interrupted and the extrusion head is swapped for the ejection of a second type of filament on the build platform. The process continues until the designed multi-material architecture is evolved.

Commercial two-elements FFF printers are available for less than EUR 10,000, and theoretically offer a resolution up to 100 µm. However, the practically achievable figures with commercially available polymers are twice this or even higher in number compared to the theoretical offered features, which quite often are not reproducible with similar printing settings. Furthermore, additional compounding, melting, mixing, or kneading cycles are required during the application of functional materials with magnetic or dielectric properties [64,151,152,153]. The round nature of the filaments, layer by layer deposition, and poor adhesion between adjacent layers is a particular limitation in single [153] as well as multi-material FFF technologies [154]. Furthermore, the limited range of materials and the thermal processing requirements corresponding to each filament create challenges in multi-material FFF 3D printing.

In contrast to FFF, vat photopolymerization-based 3D printing involves the in-situ polymerization of monomers through photoreactions. Light is illuminated according to the design of the materials, which causes localized initiation and gelation of the liquid resins. Vat photopolymerization-based technologies offer the best solution among all 3D printing technologies in view of the outstanding surface finish and printing resolution, lower equipment, and energy costs. Moreover, they are well suited for the printing of structures with multi-material properties [37,51,52,53]. The main disadvantage of vat photopolymerization-based technologies is the limited number of material classes that are available for photopolymerization. With multi-material printing, the potential is great for the combination of numerous chemical moieties into a single built material that is able to offer properties entirely different from the individual characteristics of each single component. However, in order to realize multi-material vat photopolymerization 3D printing it is essential to understand the photochemistry of the polymerization process.

## 4. Chemistry of Multi-Material 3D Printing via Vat Photopolymerization

The chemistry of vat photopolymerization technology relies on the light-triggered solidification of liquid monomers. Light-curable resins are placed in a vat, in which the polymerization reactions take place upon UV/Vis-light exposure in selective and confined areas. Monomers start to react rapidly in the presence of an appropriate photoinitiator and form crosslinked networks in a discrete manner (Figure 12). For the formation of networks with adequate properties and to achieve the required properties, it is vital to understand the structure, functionality, and underlying mechanism of the reacting monomers and photoinitiators. Utilizing the curing reaction, photoinitiators are locally activated and solidification of liquid resin takes place either layer-by-layer or point-by-point.

These reactions typically involve radical induced chain-growth polymerization of (meth)acrylates, radical induced step-growth reaction of thiol-ene and thiol-yne systems, mixed-mode thiol-acrylate polymerization, cationic polymerization of epoxides or vinyl ethers, or hybrid reaction mechanisms, which constitute dual curing networks (Figure 13). However, the rapid generation of radicals, which is followed by initiation and propagation steps in radical-induced chain-growth reactions, makes them the most often applied candidates among the available polymerization routes [141].

### 4.1. Light-Triggered Reaction in Polymers

Exploiting light to generate and crosslink polymers and to adjust their structural and functional properties has become a versatile synthesis strategy in polymer science. Owing to their numerous advantages such as low energy consumption, spatial control, and high yields, photoreactions have become indispensable for numerous industries including microelectronics, medicine, adhesives, coatings, and inks. In particular, energetically unfavorable reactions which normally would need high temperatures to overcome the activation barrier can be started at room temperature or under mild conditions simply by light exposure [79,155]. For this, irradiation of the full electromagnetic spectrum can be exploited for photoinitiation, while the choice of the correct light source is crucial to meet the required energy for the targeted photoreaction. While the energy of irradiation rises with decreasing wavelength, the penetration depth of light decreases at the same time, which often limits the use of photoreactions in thicker samples [156].

In addition, photo-reactions can be temporally controlled and are conveniently started by switching the light source on, while most processes (e.g., radical polymerization, photoisomerization, and photocyclization) can be easily stopped again by turning off the light [79].

In addition to temporal control, light as an external trigger further enables spatial control of the reactions in polymers. Control in two dimensions is accomplished by masking or patterning of the incident light. Moreover, light can be focused with optical fibers within a volume, facilitating three-dimensional control of the photoreactions. Numerous polymer processing techniques, such as photolithography and vat photopolymerization, rely on spatiotemporal control of phototriggered reactions [79,155].

Photoreactions in polymers typically follow two laws: (i) the Grotthus–Draper law, which states that only the light absorbed by a molecule is able to initiate a photoreaction, and (ii) the Stark–Einstein law, which states that each absorbed quantum of radiation initiates the reaction of one molecule [157].

Thus, photoreactions only proceed in electronically excited molecules. If the incident light has adequate energy (the spectral emission of the light source has to be overlapping with the absorption spectrum of the molecule), an absorbed photon is able to promote an electron from the illuminated molecule from the HOMO (highest occupied molecular orbital) to the LUMO (lowest unoccupied molecular orbital). The energy of an absorbed photon is described by the Planck law (Equation (1)), in which the absorbed energy (*E*) directly depends on the frequency of the photon (*v*) and Planck’s constant (*h*). The frequency is further a function of the speed of light in vacuum (*c*) and the wavelength (*λ*).
(1)E=hv=hcλ

By exposure to the appropriate wavelength, the electrons of the molecule are promoted from the ground state (S_0_) to an excited singlet state (S_n_). The excited electron then has various ways towards deactivation, which are explained in the Jablonski diagram. Following radiationless processes (internal conversion and vibration relaxation), the molecule reaches the lowest excited singlet state (S_1_). From S_1_ the molecule then returns to the ground state (S_0_) by vibration relaxation, fluorescence, or a photochemical reaction (10^−12^–10^−6^). Another possible transition with longer lifetimes (10^−7^–10 s) is from the triplet state (T_1_), which is reached by intersystem crossing (ISC). This is quantum mechanically forbidden, as the electron’s spin has to be inverted. By undergoing vibration relaxation, phosphorescence, or a chemical reaction, the molecule reverts from T_1_ back to S_0_ [157,158,159,160].

The absorption of a single illuminated molecule is described by the Beer–Lambert law (Equation (2)), in which *A* is the absorbance of irradiation within a film, *ε* is the molar extinction coefficient, and *c* is the concentration of the absorbing species. In addition, *d* corresponds to the optical path length, *I*_0_ to the incident intensity of light, and *I* to the light intensity, which is able to pass through the film [161].
(2)$$A=\varepsilon cd=-\log\left(\frac{I}{I_{0}}\right)$$

However, not each absorbed photon triggers the desired molecular process. Thus, the quantum yield (*φ*) is introduced (Equation (3)), which describes the number of times a specific event (*N_e_*) occurs per photon absorbed by the system (*N_p_*).
(3)$$\varphi =\frac{N_{e}}{N_{p}}$$

As molar absorption and quantum yield are strongly governed by the wavelength, a high absorption throughout the film is required to ensure a fast reaction rate. It should be noted that photoreactions are limited the areas of the film where light can be absorbed or intermediates can diffuse. Thus, highly absorptive, pigmented, opaque, and filled and/or thick films suffer from light intensity/rate gradients and, as a consequence, gradients in material properties.

### 4.2. Photoinitiators

Photoinitiators are molecules that are capable of generating reactive species in the form of free radicals, anions, or cations when they are exposed to ultraviolet (UV), visible (Vis), or near-infrared (NIR) light. Photoinitiators are the most essential starting units required for the majority of photopolymerization reactions. Commercial photoinitiators are available in solid or liquid form, and have to be dissolved in the resin formulations prior to irradiation.

For radical photoinitiators, there are typically two different types of initiation mechanisms that can be distinguished: Type I photoinitiators (α-cleavage), which undergo unimolecular bond cleavage to form radicals; and type II photoinitiators, which follow a bimolecular reaction yielding radicals by interaction between the excited states of the absorbing molecule and the second molecule.

For a radical photoinitiator, which forms two identical radicals, the rate of initiation, *R_i_* (Equation (4)), is directly related to the light intensity. *I*_0_, the molar absorptivity (*ε*), the radical efficiency (*f*), the quantum yield (*φ*), and the concentration of the initiator ([*I*]).
(4)$$R_{i}=2f\varphi I_{0}\left[I\right]\varepsilon$$

However, it should be noted that the equation is only valid if the film has low light attenuation through its cross-section [79].

Type I photoinitiators are typical aromatic carbonyl compounds, including benzoin and its derivatives, benzyl ketals, acetophenones, aminoalkyl phenones, O-acyl-α-oxyimino ketones, α-hydroxyalkyl ketones, and acylphosphine oxides, which generate free radicals by α-cleavage upon light exposure (Figure 13) [141,162,163]. Owing to their high quantum efficiency and reactivity, benzoin derivatives are the most commonly applied photoinitiators. However, they suffer from a short shelf life at room temperature, as the benzylic hydrogen can be easily abstracted [164]. In contrast, acylphosphine oxides benefit from an adequate thermal stability and high reactivity of the generated phosphonyl radicals [165].

Type I photoinitiators (Figure 14) are widely employed for starting crosslink reactions in vat photopolymerization 3D- printing, which mostly uses light sources emitting in the UV/Vis region. Photoinitiators with a high molar extinction coefficient and UV absorption range (λ < 400 nm) are mostly employed in SLA 3D printing [166].

The intensity and wavelength of the light required to start photocleavage reactions depends on the chemical structure of the photoinitiators. In case of Irgacure 1173 and Irgacure 651 (Table 4), the energy absorption takes place at relatively lower bands and the π to π* transition occurs in the UV range, which makes them suitable for use in SLA [159].

In the case of photoinitiators based on phosphine oxide a lower energy level of π* is exhibited, shifting the π to π* peak towards higher wavelengths and resulting in wide applications in DLP systems [161].

In contrast to type I photoinitiators, the initiation rate and curing rate of type II photoinitiators is lower, as they follow a bimolecular reaction mechanism. They require the addition of a co-initiator, which facilitates an electron transfer or hydrogen abstraction reaction in the presence of the electronically excited initiator molecular (Figure 15) [164]. In particular, light exposure of aromatic ketones, such as benzophenone, thioxanthones, benzyl, and quinones, in the presence of hydrogen donors yields ketyl radicals along with another radical obtained from the hydrogen donor [167]. For starting the polymerization/crosslinking reaction, the hydrogen donor radical is usually used. Ketyl radicals are quite stable and do not react with vinyl monomers due to steric hindrance and delocalization of unpaired electrons. However, ketyl radicals are able to terminate the polymerization reaction by forming ketyl species within the growing polymer chains [168]. Onium salts or bromo compounds are often added to the compounds to quench the ketyl radical by oxidation or bromonation and thereby prevent chain termination [164]. It should be noted that the choice of co-initiator (H-donor) is critical for a fast photoinitiation process. Tertiary amines have a higher reactivity than alcohols or ethers. While most photoinitiators absorb light in the UV range, by adding an appropriate sensitizer the absorption window can be shifted into the visible light spectral region. There are bimolecular initiator types as well, which are directly activated by visible light exposure; one example is camphorquinone in combination with an amine synergist [169].

The successful application of a photoinitiator depends on the suitability of the match between its absorption characteristics and the emitting wavelengths of the illuminating light source [91]. A list of commonly used radical photoinitiators in vat photopolymerization 3D printing is provided in Table 4.

**Table 4 polymers-14-02449-t004:** List of radical photoinitiators used in vat photopolymerization 3D printing.

Photoinitiators	Absorption Wavelength	Structure	Reference
Phenyl bis (2,4,6-trimethylbenzoyl) phosphine oxide (BAPO)	295 nm, 370 nm	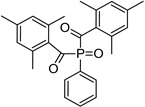	[170,171,172,173,174]
2-Hydroxy-2-methyl-1-phenyl-propan-1-one (Irgacure 1173)	245 nm, 280 nm, 331 nm	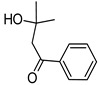	[172]
Ethyl (2,4,6-trimethylbenzoyl) phosphine oxide (Irgacure TPO-L)	275 nm, 379 nm	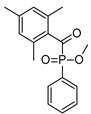	[175,176]
2,2-dimethoxy-e phenylacetohphenone (Irgacure 651)	252 nm, 340 nm	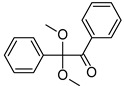	[173,177,178]
Diphenyl (2,4,6 trimethylbenzoyl) phosphine oxide (Irgacure TPO)	295 nm, 368 nm, 380 nm, 393 nm	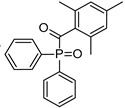	[171,179,180]
Bis (4-methoxybenzoyl) diethylgermanium (Ivocerin)	408 nm	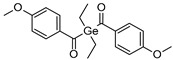	[118,175,181]
Benzophenone	253 nm	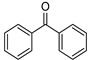	[182]
Camphorquinone	468 nm	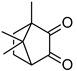	[170,183,184]
5-amino-2-benzyl-1H-benzo isoquinoline-1,3(2H)-dione (NDP2)	417 nm	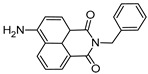	[170]
3-hydroxyflavone (3HF)	370–470 nm	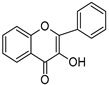	[185]

In addition to radical photoinitiators, cationic photoinitiators have gained considerable attention, as they are able to initiate polymerization reactions of epoxy and vinyl ether monomers. In the late 1970s, Crivello described the use onium salts for photochemically initiating cationic polymerizations [20,35].

Initiation relies on the formation of a Brønsted acid (H^+^), which is formed by the light-triggered decomposition of the onium salt and subsequent reaction with solvents or monomers in the formulation [37]. The chemical structure of the counter-anion governs the acidic strength of the formed acid, which increases with the size and nucleophilicity of the counter-anion [37,38].

Diaryliodonium or triarylsulfonium salts are well-known photoacid generators [186] on account of their remarkable ability to release acids when light is irradiated; selected derivatives are presented in Figure 16. Upon UV exposure, iodonium salts undergo homolytic and heterolytic cleavage reactions. In the case of heterolytic cleavage reactions, an aryl cation is generated, while in case of homolytic cleavage, an aryl radical pair or a radical cation is formed [187]. These reactive species react with neighboring monomers or solvent molecules and form strong Brønsted acids (e.g., HF), which are able to initiate cationic polymerization reactions [188].

Diaryliodonium or triarylsulfonium salts typically suffer from limited absorption (absorption maximum between 200–300 nm). One way to shift the initiation to higher wavelengths is radical-induced cationic polymerization [39]. Here, the photoacid generator is combined with a Type I or II initiator, which forms free radicals upon visible light irradiation. After they are formed, the free radicals are oxidized in the presence of the arylidonium salt and radical cations, which are able to initiate cationic reactions, are generated. Other approaches used to shift the absorption window to longer wavelengths are the use of dyes as a sensitizer [40].

### 4.3. Light-Triggered Curing Reactions Exploited in Vat Photopolymerization 3D Printing

#### 4.3.1. Chain-Growth Polymerization Reaction

Chain-growth polymerization is a reaction resulting in high molecular weight polymers at early stages of the polymerization process in which the polymer yield increases gradually with time [189]. These reactions need to be initiated by either radical or ionic initiators. The formed radicals or ionic species subsequently react with the monomers and the chains begin to grow, which is termed the initiation reaction. Propagation reactions represent the next step of the polymerization progress, during which unsaturated monomers are added one-by-one to the active centers on the growing chain [190]. The final termination of the polymerization reactions can take place either through the combination of active centers, or disproportionation reactions in which molecular rearrangements take place, exchanging the H-atom and forming a saturated and an unsaturated chain end [191].

Because chain-growth polymerization reactions take place rapidly, material properties can vary drastically. Originally low-viscous liquids are converted into glassy and highly crosslinked networks in a few seconds, which substantially impacts the polymerization performance and results in a diffusion-controlled regime and heterogenous crosslink density [192,193,194,195,196], delays in achieving equilibrium properties, and gradients in concentration and temperature.

For initiation of the polymerization reactions, aforementioned initiators need to be utilized. Under light irradiation the initiator cleaves into primary radicals, which react with the unsaturated carbon–carbon bonds in the formulation [191,196]. Hence, densely pigmented or dyed films cause gradients in light intensity and initiation rate [197]. Overall, the reaction mass transfer becomes difficult and the initiator’s efficiency declines, as primary radicals are increasingly caged and tend to combine [194,198].

The propagation step in chain-growth reactions involves the addition of monomer/oligomer radicals to the unsaturated carbon–carbon bonds, which form further radicals at the chain ends [194,199,200]. The crosslink density increases over the reaction time, leading to diffusion limitations and vitrification of the polymer. The general propagation reaction rate, *R_p_*, can be represented as (Equation (5)) [191]:(5)$$R_{p}=k_{p}\left[M\right]\left[M_{n}.\right]$$
where $k_{p}$ represents the propagation rate constant and $\left[M\right]$ and $\left[M_{n}.\right]$ represent the concentration of double bonds and total radicals, respectively. Termination of the reactions can take place with the combination of two radical species. The ideal termination kinetics does not consider the chain length dependency, radicals trapping, or heterogeneity within the evolving network, and can be presented as (Equation (6)) [191]:(6)$$R_{T}=2k_{T}{\left[M_{n}.\right]}^{2}$$
where $k_{T}$ represents the termination rate constant and $\left[M_{n}.\right]$ the overall radical concentration. The termination step is frequently limited by the diffusion-controlled regime, leading to the gel effect or Trommsdorff effect [201]. As mass transfer is limited by diffusion, termination due to the chain combination reactions becomes challenging and the concentration of primary radicals increases abruptly, resulting in an increased polymerization rate and reaction temperature. This increasing temperature causes faster radical formation, and it becomes difficult to disperse them homogeneously due to the increasing viscosity of the system [202,203,204]. This can result in the trapping of radical species within the crosslinked networks, which can hence remain in the polymer over a long service period [189,196,202,203].

##### Radical Polymerization

Free radical polymerization is a chain addition reaction in which a radical center starts and propagates the polymerization reaction. These reactions can proceed at room temperature without any extreme conditions. The reaction involves the formation of radical species by excitation and photocleavage of the photoinitiator, which starts the addition reaction [53]. These free radicals are highly reactive species and attack the monomer molecules, transmitting the active center to the attacked molecule. Hence, the reaction propagates (Figure 17) with the addition of active radical centers to other monomers (having unsaturated carbon bonds) by electron transfer throughout the activated volume and continuously increases the viscosity of resins, leading to gelation [191,196].

The reaction is terminated by immobilization of the active centers, either via combination of radical species or the deprotonation (H-transfer) of active monomers, to form stable macromolecules [191]. Radical induced chain-growth polymerization reactions are much faster than step-growth polymerization, which employs a combination of different monomers [196,199,200,205]. Acrylates, methacrylates, and vinyl monomers are the most widely applied building blocks during radical polymerization in 3D printing systems. The mechanism of the reaction is shown schematically in Figure 17.

The primary limitations of such polymer networks are observed in terms of shrinkage stresses and network heterogeneity [195,196]. Furthermore, such reactions are very sensitive to the presence of oxygen, which can react with radical species and inhibit addition reactions [164,196].

**Figure 17 polymers-14-02449-f017:**
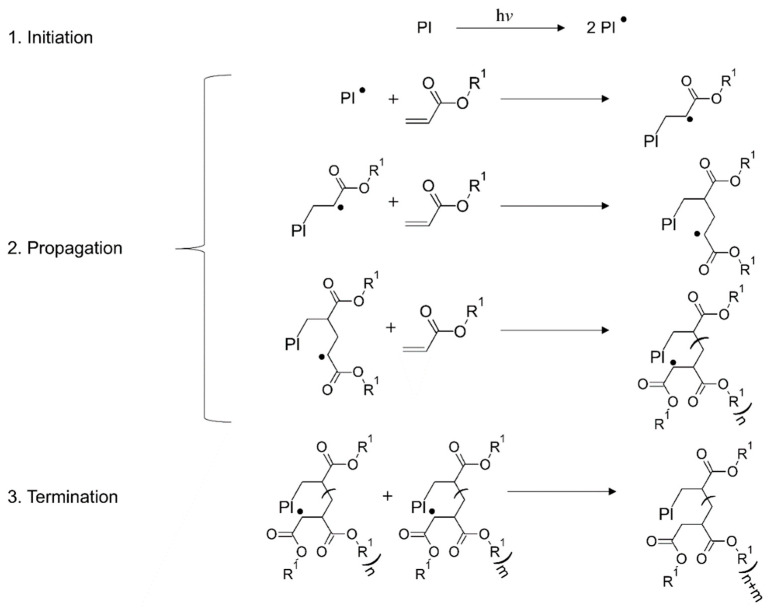
Schematics of radical-induced chain growth polymerization of acrylates. The termination stage is represented here only with a combination step. Reproduced with permission from [206]. Copyright © 2022 Elsevier.

Monomers are the building blocks of a polymer network, and greatly define the final properties of the end product. Successful completion of the polymerization reaction requires the compatibility with the 3D printing equipment, reactivity of the monomers, and appropriate absorbing characteristics of the photoinitiators and the light source. Furthermore, it is important that the printed architectures are dimensionally stable, possess the required stiffness, durability, and biocompatibility, and can withstand the application pressures, temperatures, and other environmental conditions [89,90,91,92]. Several of the widely used commercial (meth-)acrylates are presented in Figure 18.

##### Cationic Polymerization

Cationic polymerization is a class of chain-growth polymerization which involves the formation of cationic initiator species that activates the monomer by charge transfer. Cationic polymerization offers a range of advantages, including insensitivity to oxygen, higher mechanical performance, and lower shrinkage stresses in the evolved networks in comparison to radical-mediated polymerization of (meth)acrylates and vinyl monomers [207,208]. On the other hand, the reaction kinetics of cationic-initiated reactions are slower in comparison to widely adopted forms of radical polymerization. The most commonly used monomers for cationic polymerization consist of epoxies and nucleophilic vinyl monomers (Figure 19) [207,208,209].

Cationic polymerization was initially realized using cationic photoinitiators, i.e., sulfonium salts and aryl iodonium, which create reacting species as a result of illumination [188,210,211]. During epoxy ring-opening reactions, these highly reactive cationic initiators attack the adjacent epoxy rings and form polyether links via cationic ring-opening polymerization (Figure 20). The chemical reactivity of the epoxies greatly depends on their structure and functionality. Generally, cycloaliphatic epoxides have dual strained rings, and hence are particularly reactive towards ring-opening polymerization [195]. Commercial epoxies such as (3,4-epoxycyclohexane) methyl-3,4-epoxycyclohexylcarboxylate (ECC) [212,213,214,215] and bisphenol A diglycidyl ether (DGEBA) [216] are widely used in vat photopolymerization 3D printing [217,218].

As a general requirement, the liquid resins involved in SLA-based 3D printing should have a suitable viscosity < 4 Pa.s [49,220,221], as higher viscosity monomers hinder exact layer thickness during printing and increase the curing period. Furthermore, higher viscosity results in a diffusion-limited regime, as the monomers are not able to effectively disperse over the building platform during consecutive building layers, and the illuminated areas can run short of resin.

#### 4.3.2. Step-Growth Polymerization

This type of polymerization reaction typically requires initiators for starting the reaction, which they proceed between two or more different monomers with contrary functions in equimolar concentrations. In comparison to chain-growth polymerization, the molecular weight increases slowly during the early stages of the reaction and only a single type of reaction takes places repeatedly for polymer network formation [196,199,200,205].

Polymerization takes place by the reaction of functional groups from the monomer. Initially, a dimer is formed by the combination of two monomers, and dimers subsequently react with other monomers to form trimers, which combine further to form tetramers. This process proceeds until complete conversion is achieved [200,205]. The degree of polymerization ($D_{p}$) in a step-growth reaction can be expressed using Carothers’ equation (Equation (7)) for a conversion (*p*) as in [222]:(7)$$D_{p}=\frac{2}{2-pf}$$
where f represents the average number of functional groups for all kind of utilized monomers. Step-growth polymerization reactions are classified according to their reaction types, comprising polyesterification, polyamidation, polycondensation, or polyaddition reactions. However, they can take place by cycloaddition reactions, electron transfer and radical coupling reactions, and atom transfer radical addition reactions as well [191,199,200,205]. A special category of step-growth polymerization is termed click-reactions, which benefit from unique features such as: (1) performance at ambient conditions; (2) high yields; (3) insensitivity to water and oxygen; (4) regiospecificity and orthogonality with other reactions; and (5) no side reaction or easily removable side products [223,224,225,226]. Nucleophilic ring-opening reactions, copper-catalyzed azide–alkyne cycloaddition (CuAAc), Diels–Alder reactions, and thiol-ene/yne reactions are among the more well-known click reactions [223].

##### Thiol-ene and Thiol-yne Chemistry

Photopolymerization reactions based on thiol-ene or thiol-meth(acrylate) are systems which can proceed with either radical-induced step-growth polymerization or thiol Michael-addition reactions (Figure 21) [195,226]. These photoreactions are referred to as thiol-ene reactions, as the reacting group in acrylate or methacrylates is typically a -C=C moiety and presents the traits of a modular click reaction [225]. During thiol-ene free radical coupling reactions, a free radical (i.e., a thiyl radical) is added to the -C=C groups, while thiol-Michael addition reactions (TMR) involve the catalyzed addition of thiols to an electron-deficient -C=C bond. In the case of TMR, the reaction proceeds through the anti-Markonikov route, in which a nucleophilic carbon is added to an α,β-unsaturated carbonyl group through 1,4-addition reactions [12].

While the final product of both reactions can be the same, the processing methodology is entirely different [225] and a good understanding is required before correctly implementing either in a vat photopolymerization process. Thiol-Michael reactions can be initiated with a wide range of catalysts, such as Lewis acids, metals, strong bases, and organometallics [227,228], and can take place even in the dark, resulting in gelation of the liquid resins and significantly reducing their shelf life [102]. Furthermore, the fast polymerization kinetics of TMRs makes it very difficult to control the orthogonality. Henceforth, we only discuss radical-induced thiol-ene step-growth polymerization within the 3D printing domain. From a physical point of view most of the commercially available thiols are in the liquid state, which makes them convenient to dissolve and quickly introduce into photocurable resin formulations. Several commercially available alkene (Figure 22) and thiol (Figure 23) structures are presented below.

The free radical reaction of thiol-ene resins involves a series of reaction stages involving the activation of thiols in an H-abstraction reaction (Figure 24), generating thiyl radicals which are then able to attack the unsaturated -C=C bonds and shift the active radical centers by forming carbon-centered radicals. The carbon-centered radicals are able to abstract a hydrogen atom from another thiol groups, and the reaction propagates by further reaction with unsaturated carbon bonds following a step-growth mechanism [225]. By employing acrylates or methacrylates as the “ene” component, the unsaturated carbonyl bonds can undergo homopolymerization via addition reactions (chain-growth), giving rise to the mixed-mode reaction mechanism [225] represented in Figure 24.

Free radical thiol-ene reactions offer multiple advantages over conventional acrylate or methacrylate-based homopolymerization reactions. First, thiols have the ability to donate hydrogen atoms and form thiyl radicals in the presence of peroxide radicals; hence, they express insensitivity to oxygen [226,229]. Second, during thiol-ene reactions flexible thioether bonds are formed, and gelation takes place at higher reaction conversions as a result of step-growth polymerization, resulting in lower shrinkage stresses and tougher polymer networks [230,231,232].

Moreover, thiol-ene polymers present a higher level of biocompatibility compared to a pure acrylate/methacrylate polymerization system [233,234]. All of these advantages make thiol-ene or thiol-(meth)acrylate reactions one of the most promising chemistries in vat photopolymerization 3D printing, and numerous research works have adopted it in recent years [101,102,235,236,237].

Thiol addition reactions are not limited to -C=C; they are able to proceed with -C≡C groups from alkyne monomers as well [238,239,240,241]. The reaction mechanism is the same as in a thiol-ene reaction, however, it typically involves a double addition reaction with a thiyl radical, forming an additional vinyl sulfide intermediate. Thiol-yne reactions generate polymer networks with a higher crosslinking density and higher glass transition temperature compared to thiol-ene reactions [242,243,244]. Selected alkyne structures are shown in Figure 22.

#### 4.3.3. Hybrid (Interpenetrating) Polymerization

In addition to photopolymerization of single networks including vinyl monomers, (meth)acrylates, thiol-ene/yne, or epoxies, hybrid polymerization routes yielding interpenetrating networks (IPNs) have gained increased attention over the past years. IPNs are formed by the consolidation of more than one polymerization mechanism in a single resin formulation. Specifically designed to compensate for the limitations of one type of the polymerization network, the additional polymer networks are usually exploited to improve the physicochemical, thermal, or mechanical performance of the IPN. There are many examples of photocurable interpenetrating networks and selected ones are briefly described in the following section.

##### Dual Curable Networks (Photothermal Sequential Curing)

Thiol-ene/yne-based photopolymerized materials exhibit much better network homogeneity and reduced shrinkage stresses compared to pure (meth)acrylate-based polymers, as discussed earlier. However, the flexible core structure of the commercially available thiols along with the formation of inherently flexible thioether bonds significantly limits their thermal, mechanical, and physical properties [245]. This becomes particularly critical when 3D printed prototypes with higher modulus, hardness, and thermal resistance are required [246,247]. On the other hand, while pure epoxy-based networks are brittle and possess a high temperature resistance, they suffer from poor flexibility [248].

One way to benefit from the traits of both materials is to employ a combination of these monomers. Sangermano et al. developed dual curing networks consisting of a combination of thiol, acrylate, and epoxy monomers. They formulated the resins with equimolar concentrations of thiol and allyl groups and varied the concentration of epoxy monomers in the formulation. Moreover, they added 1 wt% benzophenone and 1wt% triphenylsulfonium hexafluoroantimonate as photoinitiators for the thiol-acrylate and thiol-epoxy curing reactions, respectively. The formulations were UV-irradiated, forming a mixed-mode network composed of thiol-acrylate and epoxy segments. Due to the slow curing kinetics of the cationic ring-opening reaction, a thermal post-treatment step was required in order to completely consume the epoxy monomers forming the second polymer network. They concluded that the addition of 50% epoxy monomer resulted in polymer networks with a T_g_ of 25 °C and a higher modulus, in contrast to a T_g_ of −5 °C when using pristine thiol-acrylate chemistry [247].

Morancho et al. studied hybrid networks consisting of epoxy monomers and thiol-ene resins as well. They concluded that irradiation obtained the radical-induced polymerization product, while an additional thermal curing step was required to accelerate the epoxy ring-opening reaction. The formed interpenetrating network benefited from much better thermomechanical properties in comparison to pristine thiol-ene networks [249]. Several other studies have been reported on the combination of radical and cationic photopolymerization to form IPNs using sequential curing [250,251].

Konuray et al. applied this concept in 3D printable resin formulations and developed resin formulations with varying epoxy and acrylate monomers in different concentrations. The resins were photocured during SLA 3D printing followed by thermal curing to achieve complete conversion of the epoxy monomers via cationic-induced dark curing reactions. Using these approaches, the authors developed materials with two different T_g_ values corresponding to the values of the single acrylate and epoxy networks. Moreover, varying the concentrations of constituents resulted in wide variations in T_g_ and thermomechanical properties [252]. Kuang et al. exploited a similar approach for DLP printing, using a combination of acrylate and epoxy resins for the additive manufacturing of functional devices for engineering applications [253].

Griffni et al. formulated a photothermal curing resin using a combination of light curable acrylate and thermally curable epoxy monomers, which they applied in SLA 3D printing. TPO-L was used as a radical photoinitiator, as the SLA printer contained a light source operating at 405 nm. Instead of a cationic photoinitiator, they employed classic hardeners and accelerators for the thermal curing of the epoxy monomer in a subsequent post-baking step. They demonstrated that the T_g_ could be increased from 73 °C for a pure acrylate system to 115 °C when using a blend with a 1:1 ratio of epoxy:acrylate, along with a notable increase in the storage modulus [254].

##### Dual Photo-Curing System

In addition to the dual curing cycle (i.e., photocuring followed by thermal curing), another way to form IPNs is to exploit resin formulations, which are able to form different polymer networks through a single illumination step. This idea has been exploited by using the ability of photoinitiators to initiate both radical and cationic polymerizations [248,254,255] simultaneously under UV illumination.

Various investigations have been performed on dual curing hybrid networks based on acrylate and epoxy as well as thiol-ene and thiol-epoxy networks. Jian et al. reported on a reaction system which contained a combination of thiol-acrylate and thiol-epoxy resins in a single formulation. For the photopolymerization of epoxy monomers with thiols, a photo-base generator was utilized and a radical photoinitiator was introduced in the precursor resin for triggering the thiol-acrylate reaction. An interpenetrating network was developed by simultaneously generating free radicals and a strong base by irradiating the formulation with UV wavelengths, leading to a hybrid reaction of thiols across both epoxy and acrylate groups [256]. They further reported that the T_g_ of pure thiol-acrylate networks was limited to −12.3 °C, although the interpenetrating networks formed with thiol-epoxy/thiol-acrylate (75/25 wt%) could achieve a T_g_ of 37.6 °C. Subsequently, they tuned the resin formulations in order to generate hybrid polymer networks with improved storage modulus, hardness, and thermal resistance.

Yu et al. developed hybrid epoxy–acrylate-based resins which they applied in SLA 3D printing using an illumination source at 355 nm. They adopted a similar strategy of introducing both radical and cationic photoinitiators in the formulations, which started the polymerization reactions with their respective monomers all at once upon UV irradiation. However, reaction orthogonality was not observed; instead, an interpenetrating network was developed with a single network T_g_ [248]. They reported that the cationic polymerization of epoxies could even take place in dark conditions, limiting the storage stability of the resins.

Decker et al. developed hybrid networks which underwent simultaneous curing of both acrylate and epoxy monomers upon UV exposure. They introduced applied cationic photoinitiators such as diaryliodonium hexafluorophosphate salts, which could both generate protonic acid for the cationic ring-opening reaction of the epoxy monomers and release free radicals in the presence of hydrogen donors. Thus, UV irradiation of such formulations initiated both cationic and radical polymerization reactions and successfully formed IPNs [255].

Furthermore, with recent advancements in photopolymerization and increasing applications in various fields of material science, hybrid (interpenetrating) networks have evolved as an emerging area; multiple research activities have been reported over the past several years [110,253,257,258,259,260]. The application of an appropriate reaction methodology along with a selection of appropriate monomers can further broaden the aspect of the vat photopolymerization 3D printing industry.

### 4.4. Vat Photopolymerization of Hydrogels

3D printing of hydrogels is another emerging field in the development of functional polymeric architectures. Crosslinking methodologies govern the structural development of hydrogels and maintain the mechanical integrity of their hydrophilic polymer networks [84,261,262]. Photopolymerization of hydrogels offers great spatiotemporal control [263] and flexible operation, which is vital for bioapplications such as controlled drug delivery [264]. Furthermore, the choice of monomers is crucial for the formation of hydrogels due to the associated cytotoxicity, solubility, biodegradability, biocompatibility, and stimuli response [265,266,267].

The application environment for hydrogels involves extensive interaction with water. Hence, the network should be compatible with water and exhibit suitable solubility, which often limit the application area [268,269,270]. Biologically-derived oligomers have been a great source of raw materials for hydrogel-based tissue engineering and scaffolds on account of their excellent biocompatibility and biodegradability. Gelatin is one of these materials; it has no adverse efficacy and consists of amino acid building blocks [271,272]. The presence of the cell binding motif tripeptide arginylglycylaspartic acid (RGD) primarily results in cell adhesion and diffusion along the matrix and active degradation sites, which are sensitive to metalloproteinase (MMP) [273]. Chemical modification of gelatin groups is required in order to improve the mechanical properties prior to the photopolymerization of precursors.

Thiol-ene reactions have been successfully executed together with reactions between vinyl ester-modified gelatin and thiol monomers. Vinyl ester groups are introduced by modifying gelatin with divinyl adipate [274,275]. Furthermore, gelatin can be modified by using allyl glycidyl ethers to form allyl gelatin, which can be further used in thiol-ene click-reactions. The network crosslinking degree and the related mechanical and thermal properties of the polymer networks can be controlled by varying the functionality of the thiols.

Hyaluronic acid (HA) represents another biological derivative; it consists of glucuronic acid and acetylglucosamine building blocks, and is found in connecting tissues [276]. Extracellular matrix, which contains hyaluronic acid, has great biocompatibility and supports multiple important functions including cell proliferation, migration, differentiation, and angiogenesis. The chemical modification of HA can be carried out for the formation of mechanically-tuned HA hydrogels [276]. Methacrylic anhydride is reacted with HA to yield HA–MA, which can be photopolymerized to form hydrogel networks with improved degradation times [277]. In addition, HA can be functionalized using glycidyl methacrylate to form HA–GMA, which can then be photopolymerized to form crosslinked networks with better mechanical characteristics [278]. Divinyl adipate (DVA) has been used to form HA–vinyl ester hydrogels through lipase-catalyzed transesterification reactions with hyaluronic acids [279]. Silk fibroin is another naturally occurring oligomer which has shown the potential for application in hydrogels photopolymerization. Glycidyl methacrylate is generally adopted for the modification of silk fibroin to form Sil-glycidyl methacrylate in the presence of lithium bromide [280,281]. Furthermore, 2-isocyanateoethyl methacrylate or methacrylic anhydride can be used to form methacrylated silk fibroin or methacryloyl silk protein, respectively, by reaction with silk fibroin [282].

In contrast to natural oligomers, which require reactive functionalization in order to undergo adequate photopolymerization, synthetic monomers have been utilized frequently in the photopolymerization of hydrogels. Such monomers offer low-cost operation, convenient processing, and homogenous and reproduceable results due to consistent quality [283]. Polyethylene glycol (PEG) is one of the most widely used oligomers, and has excellent biocompatibility, hydrophilicity, and insignificant cytotoxicity [284]. On account of these properties, it has been widely used in hydrogel scaffolds, drug delivery, tissue engineering, cell encapsulation, etc. Photopolymerization of PEG can conveniently control the crosslinking degree of the polymer network, consequently achieving the desired structural and mechanical properties, size of hydrogels (which can affect drug delivery), and cell interaction activities [285,286]. Similar to natural bio-based oligomers, PEG can be modified to obtain functional (meth)acrylate derivates for photopolymerization. Currently, such strategies are undertaken through the reaction between hydroxy groups from the PEG side with (meth)acryloyl chlorides to form polyethylene glycol diacrylate or dimethacrylate [287]. Furthermore, Mautner et al. synthesized vinyl ester-modified polyethylene glycol oligomers (PEGDVA) [288]. In comparison to PEDGA, vinyl ester-based PEG has a lower cytotoxicity than (meth)acrylate-based PEG, and can be easily implemented in thiol-ene-based photopolymerization systems. Polyvinyl alcohol (PVA) is another monomer with good biocompatibility and mechanical properties, and is often used in soft tissue engineering implants [289]. Similar to PEG, the successful application of PVA in photopolymerization systems requires functional modification steps involving reactions between the hydroxy groups of PVA and glycidyl methacrylate (GMA) or methacrylic anhydride in order to generate PVA-methacrylate [290].

Both the application and final properties of hydrogels are strongly dependent on the processing methodology, which governs the final architecture. The interaction of hydrogels with the environment and their diffusion, migration, proliferation and differentiation in application-specific areas requires that the materials have high resolution, crosslinking, spatial control, and structural integrity. Therefore, formation of a complex structure is very important for stimulating the complexity of the microenvironment. Photopolymerization of monomers can form high resolution hydrogels with outstanding space and time precision [291,292]. The most commonly utilized photopolymerization technologies for hydrogels include micro-patterning, stereolithography (SLA), digital light processing (DLP) 3D printing, laser printing, and two-photon absorption (TPA) [261].

## 5. Progress on Multi-Material Vat Photopolymerization 3D Printing

There are three primary strategies (Figure 25) that have been adopted by the scientific community for the development of multi-material vat photopolymerization 3D printing technology:Changing the resins during 3D printing in a sequential manner to form a multi-material structure.Printing one resin formulation while selectively activating photoreactions by changing the light source (layer-by-layer) during 3D printing.Printing one resin formulation while selectively varying the intensity of the light source (layer-by-layer) during 3D printing.

In the following sections, we briefly discuss the history of multi-material vat photopolymerization, recent works, and the pros and cons of both printing approaches.

### 5.1. Multi-Vat Photopolymerization 3D Printing

The introduction of different resins using various resin vats in a vat photopolymerization 3D printing process and fabricating the material in a chronological manner to form a multilayer structure has been well exploited by the research community. The first efforts were made by Maruo et al. in 2001; they introduced a micro-stereolithography process which was able to print with different photocurable resins. Using two polymer resin-dosing pumps in a sequential manner, they printed two different photocurable resins in a single vat and cured the materials by illumination with UV light (325 nm) layer-by-layer [293].

Later, Wicker et al. developed a multi-vat carousel assembly for multi-material stereolithography. They introduced three vats in a stereolithography setup, two for incorporating different photocurable resins and a third vat for intermittent washing and drying of the printed structure [294]. Furthermore, they demonstrated the concept of two building platforms in a multiple rotary vat setup together with a mechanical assembly to support the movement of platform and resin vat for customized multi-material printing. However, the retrofitted apparatus was limited to a building area of 4.5 inches × 4.5 inches. Their further work on multi-vat systems was extended to four rotary vat assemblies in an SLA 3D printer (Figure 26 and Figure 27) with a capability of printing resin volumes up to 9 L [57].

Choi et al. developed a simpler and non-automated multi-material system. They realized the process by intermittently removing the vat during SLA printing, withdrawing the resin, cleaning the resin vat, and introducing predetermined volumes of other resins into the vat using syringes [58].

With these strategies, researchers were able to print multi-material 3D-structures; however, the primary challenge in these concepts is the effective exchange and removal of the resin materials during the different printing steps. Furthermore, most prior multi-material vat photopolymerization works were adopted using the top-down light-scanning methodology for 3D printing, in which the platform is largely immersed into the resin volume and the 3D printing structures are built along the z-axis. Such a printing process greatly enhances the chances of contamination and resin trapping within the confined areas of 3D printed architectures, ultimately making it difficult to wash and perfectly clean the material during the printing steps.

In 2013, Zhou et al. proposed multi-material SLA printing using a bottom-up approach. With this set-up, the contact between the building platform and liquid resin was minimized and only cleaning of those parts of the structure in direct contact with the resin was required. However, their technology of multi-material 3D printing continued to rely on carousal-based vat changeover, and the cleaning process involved extensive ultra-sonification and brushing, which was not able to overcome all of the problems [295].

Matte el al. developed a multi-material DLP setup using an automated storage and retrieval system (ASRS). Instead of using rotary vat assemblies, they utilized a tower approach in which different vats were provided as vertical trays. They successfully incorporated eight resin vats into their multi-material DLP printer, and significantly improved the packing density in comparison to rotary carousel storage system. Using their developed methodology, they reduced the swapping time delays observed during vat exchange in rotary systems. Furthermore, they installed a spray cleaning unit after the exchange of vats in order to remove entrapped resin and avoid contamination of the lateral resin vats [296].

Pursuing another approach, Kowsari et al. established a multi-material DLP 3D printing setup by installing a glass plate for delivering material puddles in order to achieve faster material exchange during printing. Furthermore, they employed a novel air jet-based cleaning assembly to reduce contamination of the vats and wastage of resin while avoiding damage to the printed articles. Photocurable resins were placed in syringes and deposited on the glass plate layer-by-layer during printing. An air jet of 0.5 MPa applied in 5 s intervals was used to clean the 3D printed structures. The printing platform was adjusted above the glass plate, leaving a gap corresponding to the designed layer thickness. Multi-material structures were evolved using layer-by-layer introduction of the resins, intermittent cleaning, and visible light curing (405 nm) [66].

Ge et al. reported on multi-material printing using micro-SLA technology with different acrylate monomers and two automatic exchanging vats. They utilized a mixture of mono-functional methacrylate as linear chain builder and a difunctional methacrylate as crosslinker, and fabricated multifunctional structures with shape memory properties [61]. The process employed the automatic swapping of resin chambers after printing fixed layers of each material (Figure 28).

Keneth et al. utilized a combination of methacrylates with different glass transition temperatures using DLP 3D printing to fabricate multi-material architectures. They adopted the philosophy of print, pause, and exchange to define the steps during printing. After partially printing the initial design using one of the resins, they stopped the DLP printer and exchanged the resin vat, filled it with the second methacrylate blend formulation, and resumed the printing process for evolution of the multi-material object. Using this approach, they exploited the materials’ responses to thermal stimuli to create 3D objects which underwent localized shape memory-induced movements (Figure 29) without making modifications to their complete structure [70].

Hu et al. developed an automatic multi-vat DLP printing setup including a washing and drying chamber (Figure 30). The apparatus consisted of three different resin vats, and they employed a moving DLP projector which could illuminate the vat in all three positions, corresponding to the design. After each illumination interval, the platform was raised from the vat and transported by the motorized assembly to the washing and drying chamber, where it was washed with water and ethanol and purged before displacing it to the next resin vat [63]. Although the technology involved a time delay due to additional time consumption during the platform transfer, cleaning, and drying steps, the authors successfully 3D printed multi-material structures.

Furthermore, many other research works have been reported on multiple resin vat-based photopolymerization; these are summarized in Table 5.

In summarizing the recent progress on multiple-vat 3D printing, it can be concluded that the technology has evolved and advanced significantly, allowing for the additive manufacturing of 3D objects with multi-material properties within the last decade. However, there are immense limitations on account of the intermittent resin exchange mechanism, where extensive cleaning is required. The process is time consuming, and even automated processes cannot ensure complete cleaning of the printed structures during swapping between multiple vats. Moreover, the interlayers’ adhesion and anisotropic properties and the incompatibility of multiple polymers often results in poor material properties and premature damage to the printed objects. Hence, an optimized multi-material 3D printing technology using multiple vats while allowing for fast building of high-performance materials is yet to be realized.

### 5.2. Orthogonality-Mediated Multi-Material 3D Printing

Chemo-selective multi-material printing using vat photopolymerization has evolved as a chemical strategy to spatially control the network formation traits layer-by-layer, and presents a potential solution to the contamination and vat exchange problems encountered with multi-vat 3D printing technology. The design of the resin formulations and the polymerization strategy are the key factors that determine the orthogonality control and final chemical properties of the 3D printed articles.

Orthogonality was initially utilized in the literature to describe distinct features that could not overlap [306,307]; in 1977, it was used for the first time in chemistry to represent the selective withdrawal of protecting groups in the presence of many other groups by varying the reaction conditions [308]. Subsequently, orthogonality has been repeatedly used to illustrate selectivity in chemical interactions [309,310,311,312].

In order to achieve concurrent and independent reaction control in a single-reacting system, it is essential to induce stimuli which could drive the selective reactions without effecting the other components in the system. Various external stimuli, including heat, pH, light, redox, and electric potential, have been used for reaction initiations; among these, light is the most well-known on account of its suitable properties, including low energy consumption, spatial and temporal control, and independence from other external stimuli [313,314]. On account of light’s ability to operate independently of other external stimuli, examples of orthogonal reactions using a combination of both photo and other reaction drivers already exist for complex material fabrication [315,316], although more important for 3D printing vat photopolymerization are those systems based on the attributes of chromophores and their reactivity. One way to achieve orthogonality control in multi-reaction systems is to employ different wavelengths of light (Figure 31). This chemical reaction orthogonality coupled with the application of multi-wavelength light is frequently termed chromatic orthogonality, or λ-orthogonality in chemistry [317].

The chemical structure of the chromophores defines the capabilities of absorbing light at certain wavelengths and initiating photochemical reactions. Hence, a rational choice of a mixture of chromophores can be utilized to independently initiate selective photochemical reactions, while numerous organic synthesis conventions and post-modifications can be evolved using this reaction philosophy [318,319]. As a result, multi-wavelength light irradiation to sequentially or simultaneously initiate specific reactions broadens the prospect of performing complicated synthesis protocols, which can be challenging to drive using conventional chemical approaches. The capability to discriminately control the reaction using the wavelength can allow for materials with distinct and spatially-detailed chemical and mechanical traits [22].

Initial attempts at the development of 3D printed orthogonal vat photopolymerization protocols were made in 2019 by Schwartz et al., when they formulated a combination of acrylate and epoxy resins for DLP 3D printing (Figure 32). Radical and cationic photoinitiators were dissolved in a single resin and illuminated with visible (long wavelength) as well as UV (short wavelength) light, utilizing a multi-material actinic spatial control (MASC) strategy. Taking advantage of the reactions’ orthogonality, various multi-material 3D structures with varying spatially-induced mechanical anisotropy, swelling, and network heterogeneity were fabricated by varying the illumination wavelength [320].

Figure 33 presents a conceptual illustration of an orthogonal reaction control. Cationic photoinitiators (a mixture of triarylsulfonium salts) and radical photoinitiators (Irgacure 189) were utilized with an intrinsic absorption cut-off at 390 nm and 450 nm, respectively. In the first step, visible light was exploited to selectively activate the radical polymerization of acrylates by taking advantage of the long-wavelength absorption capability of Irgacure 819. In contrast, short-wavelength light (UV light with λ = 365 nm) was able to additionally activate triarylsulfonium salts, initiating epoxy ring-opening polymerization by forming photoacids in an orthogonal manner. Multi-material objects were fabricated using a layer-by-layer approach, with one layer illuminated with visible light and the subsequent layer with UV light. The process was repeated until the full design was 3D printed [320]. A schematic diagram of the DLP 3D printing process is shown in Figure 33.

Rossegger et al. developed a vat photopolymerization 3D printing setup using DLP technology in which they introduced a spatially activated photoacid in an orthogonal manner. The resin formulation comprised thiol and acrylate monomers along with a visible light photoinitiator. In addition, triphenylsulfonium phosphate was added as a latent transesterification catalyst (photoacid) with the capacity to release Brønsted acid under UV irradiation. The resin formulation was DLP 3D printed with both visible light (405 nm) and UV light (365 nm) projectors, as shown in Figure 34. Cured structures were evolved by illuminating the resin layer-by-layer with visible light and UV light in an alternating manner. In this way, the photoacid was activated orthogonally in the desired layers, which was exploited as the catalyst to initiate dynamic exchange reactions at elevated temperatures [237].

### 5.3. Grayscale Photopolymerization 3D Printing

Grayscale illumination is a process for locally controlling light doses during 3D printing, leading to a spatially customized crosslink density in the polymer network. The technology employs a single vat photopolymerization unit, although it could be based on multiple vat units as well.

The core methodology is based on grayscale processing, in which images are scanned in a monochromic light setting in order to induce location-specific properties within the network. The designed articles are sliced into images corresponding to each illuminating layer and materials are built in a layer-by-layer approach. Mathematical software tools are then utilized to process the individual images and set the grayscale distribution according to the preferred material properties. Grayscale images are derived from the normalization of red-green-blue (RGB) values to determine grayscale percentages and classified from 100% (dark scale) to 0% (full intensity). These programmed images with grayscale impressions are transmitted to the light projection modules for illumination and vat photopolymerization. Higher grayscale levels result in a lower crosslink density, and hence a lower modulus of the materials. Empirical relationships can be further derived by determining the conversion of reactive functional groups with light illumination [321].

Kuang et al. formed functionally graded materials with a wide range of mechanical properties using grayscale vat photopolymerization. Their technology of grayscale printing was based on digital light processing. As photocurable resins, a combination of mono-functional and di-functional acrylates were selected along with a methacrylate bearing epoxy moieties, amine crosslinkers and photoinitiator. Initially, acrylates in the resin formulation were allowed to radically photopolymerize, leading to a polymer network formation which could fix the designed structure [321]. Regions that were underexposed during greyscale polymerization resulted in unreacted monomers, potentially resulting in poor mechanical properties [322,323,324,325]. A second thermal curing step was introduced to convert the unreacted monomer and improve the mechanical properties of the underexposed greyscale regions. The primary contribution to thermal curing was observed from the crosslinking reaction of diamine with acrylate and epoxide monomers [321]. The thermal curing of the articles was carried out at below 120 °C, effectively avoiding self-initiation of acrylate homopolymerization [326]. Materials formed with higher grayscale (93%) factors were typically soft and had a Young’s modulus up to 1.4 MPa, while stiffer material regions were formed by adjusting the grayscale to 0% and had a Young’s modulus of up to 1.2 GPa. A wide range of graded materials were fabricated using this approach, with a T_g_ varying between 14 to 68 °C [321].

Wu et al. further devised grayscale light patterns to control the light intensity of light projection. DLP vat photopolymerization technology was adopted to develop spatially varied materials by illuminating each layer for the same amount of time and employing digital mask patterns, which ultimately varied the crosslinking density distribution. The fundamental principle was based on the fact that dark scaling corresponds to lower light intensity, and thus results in materials with lower crosslinking density. Mathematical integration software programs such as Matlab and Solidworks were employed to obtain the grayscale values of each pixel of the slicing image. Grayscale values ranged between 0–255, from a completely black to a completely white scale, respectively. These grayscales were fed into a digital micromirror device projector, which transmitted the grayscale patterns into the corresponding light intensities in such a way that completely black patterns (0 grayscale) led to 0% light illumination and completely white patterns (255 grayscale) led to 100% light intensity. Hence, the crosslinking degree was spatially controlled by transforming grayscale patterns into a light intensity distribution in each layer while using resin formulations containing conventional acrylates and methacrylates [325].

Muralidharan et al. employed a combination of thiol-acrylate and thiol-ene based hydrogels for developing composite materials by employing grayscale SLA technology. For pursuing the grayscale illumination strategy, the resin formulations were experimentally studied in order to determine the relationships between the illumination intensity, illumination time, and reaction conversions. The reaction conversions corresponded with the crosslinking density, which was verified by swelling test measurements. In this way, regions of soft and stiff materials were successfully integrated in a composite structure by employing grayscale impressions [34].

## 6. Applications of Multi-Material Vat Photopolymerization 3D Printing

Vat polymerization 3D printing technologies have clearly presented themselves as an outstanding competitor among all the various additive manufacturing technologies on account of their high printing resolution and building speed and of the surface finish of their final products. The ease of upscaling and simplicity of combining various chemistries has led to complex 3D designs which can be quickly realized as real objects at lower costs. In this section, we discuss the applications of vat photopolymerization-based multi-material 3D printing in various materials-related fields.

### 6.1. Bio-Mimicking by Combining Soft and Hard Segments

The existence of natural high-performance biological structures and evolution over the past hundreds of years has been a great motivation for the design and fabrication of materials that could mimic natural bodies. The complex structural geometry of bio-inspired architectures has surpassed the complexity limits that were achievable using additive manufacturing with single materials. Mimicking the architectures of natural bodies, such as bones and nacre, which comprise two different materials (as in a brick-and mortar-design, that is, the stiff and the soft part, respectively), in order to achieve higher strength and energy-dissipation characteristics, is nearly impossible via fabrication using single-material 3D printing technologies. Multi-material 3D printing could revolutionize material developments for replicating such multi-functional and multiscale natural entities.

Schwartz et al. worked on multi-material vat photopolymerization and printed polymer structures which mimic an anisotropic human hand. Applying dual-wavelength photopolymerization, they used a stiff polymer to form the internal bone structure along with a soft polymer wrapped around the rigid structure. A combination of acrylate and epoxy resin was formulated and orthogonally cured with visible light to yield a soft acrylate network, while the stiff epoxy–acrylate IPN was formed using cationic polymerization (UV illumination) [320]. Anisotropic compression in multi-material architectures was attained through well-defined soft and hard segments. Figure 35 shows the resulting multi-material DLP 3D printed hand structure.

Theses anisotropic properties can be tuned to develop a different set of multi-material structures by varying the contents of acrylate and epoxy in the formulation. Furthermore, the capability of such multi-material systems to spatially control the strength and anisotropic features by controlling the content and location of soft and rigid polymers is a technically relevant aspect in bioinspired materials.

Furthermore, metamaterials with a negative Poisson’s ratio have been developed using multi-material vat photopolymerization. Such materials have the capability of becoming thicker perpendicular to the applied force in stretched conditions. This property is particularly useful in preventing of fractures and impact resonance. Chen et al. fabricated metamaterial lattices consisting of spatially distributed rigidity over the material by employing an automatic resin switching process. They employed resins comprising a stiff and low-molecular mass polyfunctional tri-acrylate and a very flexible high-molecular mass oligomeric polyether(meth)acrylate in order to visualize hard and soft segments. By varying the content of these monomers, they could tune the modulus and Poisson’s ratio in microlattices over a broad range (Figure 36) [327].

Peterson et al. developed materials by spatially controlling the crosslinking density with greyscale DLP technology. Multi-material lattices and trusses were fabricated with distinctive mechanical features by 3D printing using commercially available acrylate-based resins. Octet trusses with lower crosslinking density at the joints and higher crosslinking density at the beams were produced, which significantly improved the mechanical properties of trusses in comparison to constant greyscale printing [322]. Kuang et al. developed compression lattices and made use of greyscale DLP 3D printing to tune the deformations, negative Poisson’s ratio, and structural anisotropy in their work [321].

### 6.2. Bio-Medical Applications

Multi-material vat photopolymerization 3D printing has been a very prominent technology for mimicking biological structures and synthetic tissue architectures. Owing to the ability of multi-materials to incorporate different properties into a single complex, numerous cell types and natural tissues can be developed on a reasonable time scale.

Wu et al. developed a customized vat-interchanging SLA system for the fabrication of a bi-phase osteochondral scaffold for implantation in a goat’s knee joint. The scaffold was developed using poly(ethylene glycol) diacrylate (PEGDA) to act as the cartilage scaffold in the multi-material and a ceramic beta-tricalcium phosphate (β-TCP) as the bone tissue scaffold [328]. Furthermore, tissue models have been developed using multi-material vat photopolymerization technology. PEG and hyaluronic acid derivates were implemented as degradable scaffolds in a cell-laden biopolymer to obtain channels in vascular model and liver tissues [329,330]. The same process exploited DLP 3D printing with the exchange of materials in the resin vat (Figure 37).

Moreover, maskless microfluidics-enabled multi-material 3D printers using DLP technology have been used to fabricate various models of tissue geometry, such as muscle strips, angiogenesis, and muscular skeleton joints, employing polymer precursor resins, including poly(ethylene glycol) diacrylate (PEGDA) and gelatin methacryloyl (GelMA) in different compositions (Figure 38) [331].

Moreover, many other studies have described the use of stereolithography with exchanging resin systems for additive manufacturing of biocompatible multi-material polymers such as polyethylene glycol dimethacrylate and polyethylene glycol diacrylate [65,332,333,334,335,336,337,338,339,340,341,342,343,344,345].

Multi-material 3D printed polypills have been developed in recent years using SLA technology with exchange vats during fabrication. Drugs are introduced into bio-compatible photocurable resins in required doses and illuminated for curing and structuring [332,333]. This can support patients’ efforts to undertake defined volumes of medicine or multiple medicines in a single shot, as well as reducing non-adherence. Figure 39 shows a multi-material polypill developed in PEGDA using a blend of six medical drugs (naproxen, aspirin, paracetamol, caffeine, chloramphenicol, and prednisolone) and printed layer-by-layer using each blend formulation [332]. Furthermore, other applications of multi-material vat photopolymerization in bio-medical industry are summarized in Table 6.

### 6.3. Stimuli-Responsive Behavior and Robotic Actuators

Stimuli-responsive behavior describes the tendency of materials to exhibit physicochemical transitions as a result of changes in their environment [346]. Multi-material 3D printed polymers could offer exceptional traits of stimuli responsiveness thanks to their inherent structural features. The presence of two or more polymer networks in a composite material could generate different responses based on the driving force, and numerous works in this direction have been reported.

Kuang et al. adapted greyscale DLP 3D printing to create multi-material polymer networks with a T_g_ ranging from 14–68 °C within a single multi-material structure by spatially varying the illumination intensity. The authors made use of a broad temperature range and developed shape memory strategies by actuating the 3D printed parts at different temperatures [321]. Furthermore, Wu et al. developed structures based on PEGDA, butyl acrylate, and butyl methacrylate, comprising regions with varying crosslinking density. These domains reacted to solvent absorption and swelling to varying extents, resulting in spatially-controlled shrinkage and reversible bending deformations [325].

Keneth et al. exploited DLP 3D printing to prepare multi-material structures with tailored T_g_ values, which were used to induce motions with thermal or photo-stimuli. Polycaprolactone methacrylate (PCLMA) and a mixture of *N*-vinylcaprolactam and PCLMA were introduced as two different polymer precursors in two different resin formulations. Multi-material boxes were fabricated by initially printing the structures with the first formulation, pausing the printer, and adding the second resin in the vat for printing with a resin mixture. The shape memory-based activations were carried out by placing the 3D printed box in a hot water bath (Figure 40). Pure PCLMA was activated at 52 °C, which resulted in the box-top lid opening, while mixed resin was activated at 58 °C, resulting in closure of the inset box lid [70].

Ge et al. developed multi-materials with spatially-controlled shape memory properties. They synthesized methacrylate-based copolymers and varied the reaction constituents and compositions, obtaining photopolymer networks with T_g_ values varying between 50 and 180 °C. In addition, the modulus could be adjusted between 1 MPa and 100 MPa. SLA 3D printing of multi-material structures was carried out via automatic swapping of the resin vats according to the sliced 3D design [61]. Various designs of grippers were 3D printed and programming of the transitions over a wide temperature range by controlling resin properties was presented (Figure 41).

Furthermore, they developed a multi-material flower structure, with the inner petals having a T_g_ of 56 °C and the outer petals having a T_g_ of 43 °C. All of the petals were manually closed at 70 °C and then cooled down to 20 °C to fix the structure (programming step). Figure 42a presents the structure of the petals after the cooling stage. The temperature was then increased up to 50 °C to allow the outer petals to expand, resulting in blooming (Figure 42b), while the internal petal remained closed. After the temperature was increased to 70 °C, the inner petals opened up to recover the original shape of the flower [61].

Thrasher et al. designed multi-material pneumatic grippers, a Gyroid lattice, and an Octet truss and fabricated them by applying multi-material vat photopolymerization. During the printing process they employed several photocurable resins: a low-modulus flexible, high-modulus flexible, and hard material, in a DLP 3D printing setup. The designed number of layers was printed and then the printing was paused, followed by replacement of the photocurable resin in the vat [347]. Using this approach, the multi-material gripper shown in Figure 43 was developed.

Rossegger et al. took advantage of two transition temperatures in the developed multi-material structures using the dual wavelength vat photopolymerization approach. While the thiol-acrylate network was formed at 405, UV irradiation at 365 nm locally activated a transesterification catalyst, resulting in dynamic exchange reactions at elevated temperatures. Above the vitrification temperature (T_v_), the UV-irradiated domains become malleable, which can be exploited to locally change the shape memory properties. Figure 44a presents a bar, one half of which was printed with visible light and the other printed with UV light (and hence contained activated Brønsted acid) [237].

During the programming step, the temperature of the gripper (permanent shape) was increased above T_v_ in a U-shaped mold, which turned the material from both ends, while the shape of the bar was fixed by cooling it down to 10 °C (Figure 44b, shape II). Further heating of the material above T_g_ resulted in a special shape in which the UV-activated side remained bent, while the visible light-activated side regressed back to the permanent bar shape (Figure 44b, shape III).

A similar experiment was repeated with a gripper (Figure 44c), in which two arms were printed with visible light and the other two with UV light (Figure 44a). After the programming step, only the localized motion of the visible light-exposed arms was observed, while the UV-activated arms remained locked in programmed positions on account of the Brønsted acid-catalyzed topological rearrangements (Figure 44c, shape III) [237].

Furthermore, several other applications of stimuli-responsive materials formed by multi-material vat photopolymerization have been reported in the recent years; these are summarized in Table 7.

### 6.4. Multi-Material Structures with Self-Healing Functionality

Vitrimers are a class of materials with intrinsic dynamic bonds within the chemically crosslinked networks [350]. These dynamic bond exchanges are thermally activated, resulting in microscopic reflow of the material under the application of certain forces while maintaining the structural integrity of the cured architectures. Vitrimers are known for their ability to self-heal and recycle under the application of heat and mechanical stimuli [101,102,351,352].

Pure acrylate and thiol-acrylate-based 3D printed vitrimers have been developed using vat photopolymerization in recent years, comprising transesterification-based dynamic reactions [101,102,206,226,351]. Rossegger et al. developed vitrimeric systems using dual-wavelength vat photopolymerization technology. Through orthogonal activation of Brønsted acid catalysts in UV-exposed regions, transesterification reactions could be locally activated within the 3D printed structures, which were then further exploited for thermo-activated healing. The quantification of dynamic response is carried out in terms of stress-relaxation experiments, during which the time decay of the constantly applied stress is measured. Figure 45a represents the dynamic responses obtained by incorporation of photoacid generator in multi-materials formed using a vat photopolymerization 3D printing process [237].

All UV-activated multi-material samples represent reasonable stress relaxation properties, however, the non-UV-activated multi-materials and the samples without photoacid generators exhibit very slow stress relaxation behavior, confirming the exceptional activity of dynamic reactions in UV-irradiated domains. Furthermore, the relaxation time behavior over various temperatures follows a linear trend (Figure 45b), which is a characteristic feature of vitrimeric materials [353,354], hence confirming the great prospects of orthogonally-fabricated multi-materials in the field of vitrimers.

## 7. Further Aspects of Multi-Material Vat Photopolymerization 3D Printing

The layer-by-layer formation of materials via additive manufacturing processes has been commercially available for more than 25 years. In this section, we discuss several technical and economic aspects of multi-material vat photopolymerization 3D printing processes.

### 7.1. Postprocessing of Multi-Material Vat Photopolymerization 3D Printing

Most multi-material vat photopolymerization approaches rely on radical-based photopolymerization of acrylates, thiol-ene or thiol-yne reactions, and ring-opening polymerization of epoxy monomers or thiol-epoxy reactions. Post-curing of the photocured materials is often carried out, with the goal of either completely polymerizing the polymer network or improving the material properties to withstand certain application conditions. Generally, photopolymerization reaction conversions are a function of time, illumination wavelength, and light intensity. Although the reactions generally proceed quickly in such polymerization mechanisms, the conversion of certain monomers can be limited, and may not reach 100% during the photopolymerization process. In such situations, a post-thermal/light treatment step is required in order to fully crosslink the networks. Printed materials are removed for the 3D printer platform and washed with solvents (usually ethanol or propanol) to remove the unreacted monomers from the surface and clean the printed articles prior to thermal treatment.

Sangermano et al. studied the photocuring of a hybrid thiol-acrylate-epoxy resin system and photocured materials with UV irradiation. It was observed that the ring-opening reaction of epoxy groups proceeded at a much slower rate, and required a second thermal treatment process for high conversion of the epoxy monomers [247]. Morancho et al. studied thiol-ene and epoxy blend networks, which suffered from low thermo-mechanical properties until the thermal curing step was performed and epoxy polymerization took place [249]. Konuray et al. developed an SLA 3D printable resin containing epoxy and acrylate monomers. Epoxy curing was carried out in a thermal post-curing step, which was visible as two distinct glass transition temperatures [252].

Schwartz et al. studied the impact of thermal treatment on the network, curing, and mechanical properties of orthogonally-formed multi-materials using epoxy and acrylate-based resins. Multi-materials were DLP 3D printed by illuminating them with visible and UV light in a sequential manner, causing them to undergo radical and cationic polymerization, respectively. In order to study the network properties and the impact of thermal treatment on individual materials, UV- and visible light-cured samples were thermally postprocessed at 60 °C; it was observed that the stiffness of the materials increased significantly for the UV-cured samples only (Figure 46a), while the stiffness remained almost unchanged for the visible light-cured samples. This observation coincided with the fact that UV curing resulted in unreacted epoxy monomers becoming trapped within the 3D printed structures, meaning that thermal curing caused further ring-opening reactions and an overall increase in crosslinking density [320]. The results of shore-A hardness tests for the UV- and visible light-cured networks are exhibited in Figure 46b. The results further reveal that UV curing resulted in increased crosslinking with prolonged photo-illumination time, while thermal curing (a post-treatment step) resulted in conversion of the trapped unreacted monomers, forming a stiffer and harder network. No significant impact of thermal treatment was observed on visible light-cured networks.

We investigated the effect of thermal treatment on DLP 3D printed structures from thiol-acrylate resins [101]. The conversion of acrylates reached more than 90% during visible light curing in DLP 3D printing; meanwhile, the conversion of thiol was limited to 60%. Thermal treatment of the samples resulted in further curing of the trapped unreacted monomers, and the conversions increased after thermal post-curing, which led to a higher crosslinking density. Furthermore, hydrogen bonds were formed within the network during thermal treatment, which increased the stiffness of the network and contributed to enhanced crosslink density (Figure 47).

The impact of thermal post-curing on the mechanical properties of thiol-acrylate DLP 3D printed materials was investigated (Figure 48a,b), and it was observed that thermal annealing significantly increased the mechanical properties of DLP 3D printed materials (Figure 48b). The formation of hydrogen bonds and the additional conversion reactions driven by heat treatment governed structural rearrangements and bond formations that significantly increased the mechanical properties [101].

In another study from our group, we studied the impact of thermal treatment on the mechanical properties of DLP 3D printed resins and found that the strain at break and ultimate tensile stress improved from 23% to 47% and 0.26 to 3.1 MPa, respectively, as a result of thermal post-curing [102]. Furthermore, the glass transition of the vat photopolymerized structures increased from 0 °C to 20 °C with thermal treatment at 180 °C for 4 h (Figure 49), which was attributed to the formation of additional hydrogen bonds and crosslinking reactions of trapped unreacted monomers.

### 7.2. Thermal and Mechanical Aspects of 3D Objects Fabricated via Multi-Material Vat Photopolymerization

The thermal and mechanical properties of 3D printed multi-material structures strongly rely on the compatibility and multilayer interactions of the applied materials. As multi-material vat photopolymerization 3D printing involves the combination of different resins or networks (either via resin exchange, orthogonally-mediated reaction mechanisms, or grayscaling), it is vital to characterize the interaction of composite materials at the micro-level.

In contrast to vat photopolymerization, the mechanical properties of extrusion-based processes are controlled by the thermal dynamics experienced by the filaments during the extrusion, deposition, and cooling processes [355,356]. Filaments being extruded via thermal processes have to pass through four different stages of processing in the sintering, crystallization, glass transition phase, and shrinkage steps [45,356]. The interlayer adhesion strength of the material layers being extruded depends heavily on these processing stages, which often results in limited mechanical characteristics. The improvement of mechanical properties in such materials requires fine temperature control, knowledge of the filaments’ prior processing, chemistry, temperature dependence of viscosity, and good understanding of their material chemistry and thermal stability [356,357,358,359]. In addition, thermal shrinkage of the materials and their poor surface finish, which may require expensive and elaborate post-processing steps, limit the resolution of the materials [42,360]. In contrast, vat photopolymerization involves the input of liquid resins which are conveniently processable, mostly at room temperature; the reacting monomers form polymer networks at a molecular level, which can significantly improve the multilayer interactions, overall surface finish, material resolution, precise layer thickness control, and mechanical properties. However, shrinkage is an issue for 3D objects formed by free radical photopolymerization of acrylate monomers.

Ge et al. prepared multi-material structures by employing an automatic resin vat exchange assembly in an SLA 3D printer to form shape memory materials. In addition, they characterized the thermomechanical properties and the interface adhesion strength of the multi-material structures, which are presented in Figure 50.

Material A, comprising a mixture of 50% benzyl methacrylate (B) and 50% poly(ethylene glycol) dimethacrylate (P550), had a glass transition temperature of 32 °C, while material B, composed of 90% benzyl methacrylate and 10% bisphenol A ethoxylate dimethacrylate (BPA), had a glass transition temperature of 56 °C. The multi-material structures formed using both resins had two glass transition temperatures at 33 °C and 60 °C (Figure 50a), which confirms great integration and polymerization control of individual networks. Furthermore, the adhesion strength of the multi-material structures was measured through uniaxial tensile testing of 3D printed strips (Figure 50b). For the purpose of comparison single-resin versions of A and B were 3D printed separately and tensile tests were performed under similar settings (Figure 50b). The authors reported that the multi-material structure did not fail at the interface, rather cohesively within the domain formed by resin A. Thus, the results show the great compatibility and adhesion of the photopolymer networks formed during multi-material 3D printing [61].

Furthermore, the tensile properties of both the individual materials A and B and the multi-material samples were determined (Table 8). It was found that the multi-material structures had properties lying between the upper and lower limits defined by the single materials A and B.

Schwartz et al. prepared multi-material structures consisting of pillars of a stiffer UV-cured network embedded within a matrix composed of a soft visible light-cured network (Figure 51a,b). Compression tests were performed on the individual UV-cured resin and visible light-cured resins and the performance was compared with the multi-material structures. It was observed that UV-cured networks (Figure 51c, black) reached a maximum stress limit (500N load cell) at very low strain values (~10%), while visible light irradiation yielded flexible structures (Figure 51c, red) with very high compression strain values (up to ~90%) without reaching the load cell limit (no fracture onset was physically observed) [320].

In contrast, the multi-material structures exhibited anisotropic properties and their compression behavior was orientation-dependent. Compression testing of the multi-material structures along the x-axis (Figure 51c, yellow) was initially similar to the softer network; however, after 59% compression strain, the stress was distributed evenly across the stiffer and softer networks, resulting in the fracture of the pillars at around 75% strain. Compression of along the z-axis (Figure 51c, blue) resulted in bilinear behavior, and the compression of the stiff pillars led to early onset of failure (~50% strain) [320].

### 7.3. Economical Aspects of Vat Photopolymerization 3D Printing

The beginning of a business with the implementation of a new technology is a very challenging process, and often requires complete knowledge of capital and running costs along with a clear picture of the potential supply chain. 3D printing, specifically vat photopolymerization 3D printing, can be effectively used to increase the economic values of materials thanks to its exceptional product resolution, surface quality, multifunctionality, and cost-effective operation. Other factors, such as the freedom to manufacture delicate materials in-house irrespective of the risks generated by depending on raw material suppliers and supply chain channels, and the protection of privacy, while hard to quantify, always remain at the center of economic evaluation [35]. Owing to the tool-less nature of vat photopolymerization 3D printing technology, it can effectively reduce manufacturing costs for processes involving high customization and complexity when smaller production volumes are required [361]. Photopolymerization has been gaining market ground in rapid prototyping, dental implants, and biomedical applications as a real-time processing technique, leading to a reduction in time to market and time to profit strategy [35]. Prospective growth and future milestones can now be attained by direct manufacturing of consumer goods, industrial and domestics parts, equipment housings, machine tools, etc.

The cost of any vat photopolymerization technology involves the cost of 3D printers (equipment), resin raw materials (resin precursors), cleaning chemicals (ethanol, isopropanol, acetone, etc.), thermal post-curing equipment, energy costs (electricity), and 3D design costs (Figure 52).

Commercial 3D printers based on vat photopolymerization technology are available in different sizes and production capacities. The most widely used technologies, SLA and DLP 3D printers, are available from Formslab, Anycubic, and many other commercial brands. Single-material 3D printers based on SLA technology are available in the price range of USD 3000, while larger 3D printers with a larger building volume (33.5 × 20 × 30 cm) can cost more than USD 11,000 [362]. In contrast to SLA, the DLP technology printers offered by Anycubic are available in wide range of sizes and production capacities. A small lab-scale DLP 3D printer with a building capacity of 13 × 8 × 16.5 cm (L × W × H) costs only around USD 200, while larger DLP 3D printers (30 × 16 × 30 cm, LWH) are available in the range of USD 1100 [363].

The operating (electricity) and raw materials (resin and cleaning liquids) costs can vary depending on the production targets and the size of the objects which are being 3D printed, as well as on the volume of materials procurement, as the chemical industry relies heavily on bulk transfers, which significantly benefits the consumers as well in the final economic aspects.

Multi-material vat photopolymerization printers made of vat exchange mechanisms are derivates of single-material 3D printers, mostly customized and built in-house, with additional arrangements of vats, transferring assembly, and intermediate cleaning tools. Orthogonally-mediated 3D printing systems require much more sophisticated systems and need a great degree of control for successful multi-material formation. Schwartz et al. developed a customized 3D printer for the formation of multi-material structures by integrating illumination lamps into the printing setup [320]. Such dual-wavelength 3D printers are currently on developmental scales and need advancement before being ready for commercializing and large-scale manufacturing. Typical standard and engineering resins utilized in vat photopolymerization 3D printing are commercially available in the price range of USD 149–200/L [362], while customized 3D printing applications require synthetic resins, which can cost even more based on the synthesis complexity and cost of the reacting blocks. However, the unmatched surface quality, customization possibilities, and building resolution of vat photopolymerization surpasses the technological hurdles, and makes it one of the most competitive additive manufacturing technologies.

## 8. Outlook and Prospects

Additive manufacturing of materials has presented great potential over the past decade, and has been adopted under many technology subdivisions by material researchers and the industrial community. The desire of mankind to mimic nature and to create materials that could offer multifunctional traits such as human bones, soft matter with supporting structure, and natural bodies has given rise to multi-materials and many new additive manufacturing technologies. Vat photopolymerization has become a popular technology among 3D printing technologies due to its excellent printing resolution, dimensional stability, high-quality surface finish, and fast processing, although the technology seriously suffers from the limited functionalities of the resins and materials that can be processed and photo-cured for vat photopolymerization [53]. While customized resins can be synthesized at research scales to achieve the required traits, the process becomes time- and energy-intensive, and incurs a higher cost in terms of the required raw materials and synthesis assemblies.

Hence, on the one hand it is important to discover new resin material chemistries, while on the other the focus should be placed on utilizing those which are currently available in an effective way and making the most out of them. One way to achieve this is the multi-material approach, which can take the advantage of what materials are on hand or commercially available.

Multi-material 3D photopolymerization offers the advantages of combined material chemistries along with the cumulative advantages of vat polymerization; however, suitable strategies need to be built for developing the process. The capability to freely integrate multiple materials having diverse mechanical, biological, optical, and electrical properties could represent the most significant aspect of the current decade in terms of making 3D printing more competitive.

However, the actual level of multi-material vat photopolymerization-based materials are currently simplistic and mostly based on multi-vat processes. Such systems result in broadly contaminated products and suffer from the need for extensive cleaning and time delays during inter-vat changeover. In contrast, the chemical advancements in orthogonality-mediated multi-materials are limited, and need further developments in order to advance the complexity and integration levels of the materials industry. Application of two wavelengths for curing monomers is an attractive feature for network functionalization in photo-chemistry.

Selective and orthogonal photoreactions in the presence of two or more photochromes could lead to spatially and temporally controlled multi-materials with superior levels of integrity and interlayer interactions. Multi-wavelength light sources, which can spatially activate the distributed chromophores, are able to form different networks with distinctive photo-illumination.

The combination of epoxy and acrylate monomers in combination with photo-selective polymerization, i.e., cationic and radical polymerization using dual-wavelength vat photopolymerization, has already been used to fabricate multi-material networks at different wavelengths. As a future prospect, we believe that the combination of photoreactive monomers, such as (meth)acrylates, epoxies, vinyl monomers, or alkynes, with the cycloaddition reaction, such as photo-favored [2π + 2π] and [4π + 4π] mechanisms, could develop multi-materials with orthogonality-mediated network formations.

Photocuring of polymeric resins in photoresists was one of the earliest technologies to be developed for micro-structures and surface treatments [364,365]. While vat photopolymerization technologies were first reported in late 20th century, they employed similar principles that originated from photoresists. The formation of photoresists relies on analogous chemical ingredients as vat photopolymerization technology does today, with the difference being transmission from smaller to macro-level features development. The latest trends and technological advancements have diverted interests from simplistic photoresists and vat photopolymerization technologies to dual-wavelength photocuring systems, as they offer the possibility of locally forming different polymer networks [312,366,367]; various studies have been presented in recent years in this direction [368,369].

Photocycloaddition reactions offer the capability to locally switch material characteristics reversibly by wavelength-selective network formation and cleavage reactions. Photo-dimers can form at typical long wavelengths >300 nm (low energy), while cycle-reversion can occur when exposed with short-wavelength light of <300 nm (high energy). Furthermore, [2π + 2π] cycloaddition reactions represent a class of photoreactions employing styrylpyrenes, coumarins, and thymins, and possess the attractive traits of selective bonding and debonding under wavelength-selective illumination. These monomers find special applications in photo-induced self-healing, photo-triggered shape memory, and reversible photoresists.

Coumarin represents another photochrome which finds applications in micro- and macromolecular applications [370,371,372,373]. Such monomers undergo [2π + 2π] cycloaddition reactions with UV illumination (350 nm), forming cyclobutene rings (Figure 53) [373], while irradiation with low-wavelength light (254 nm) can photocleave the crosslinks in a reversible manner [370,374].

Anthracene monomers are well-known in photochemistry on account of their ability to undergo [4π + 4π] cycloaddition reactions under photo-illumination in the range of UV-visible light, forming dimers [376,377,378,379,380]. These monomers possess fluorescence properties, which make them distinct from the dimerized state, where the fluorescence is fully lost. Barner-Kowollik et al. took advantage of the photocuring properties of acrylates bearing anthracene moieties to present a multi-material formation strategy using direct laser writing (DLW). With DLW, they formed networks using radical polymerization of acrylates, which served the purpose of cleaving partially-existing anthracene dimers. Further illumination with LED light of 415 nm resulted in a cycloaddition reaction of anthracene, forming dimers via [4π + 4π] cycloaddition reactions. The higher laser illumination times during DLW resulted in higher fluorescence generation as a result of anthracene dimer cleavage reactions [381].

A promising area of chemistry relying on chromophores in wavelength-mediated photo-activation is the *o*-nitrobenzyl ether family, which have the ability to undergo a cleavage reaction in the UV_A_ region and are transparent to the visible light spectrum [382]. Such chromophores find applications in photo-resists [383] and local drug release [384,385]. Photolabile *o*-nitrobenzyl esters have been implemented in thiol-click photo reactions for wavelength-based modifications of surface properties, reactivity, and thermomechanical properties [386,387]. Radl and co-workers prepared switchable photoresists by introducing *o*-nitrobenzyl ester moieties in visible light-curable thiol-ene networks. While the network was formed by exposure at 405 nm, the covalent link could be selectively cleaved in a subsequent UV exposure step [388]. Barner-Kowollik et al. synthesized an acrylate-based monomer with *o*-nitrobenzyl ether chromophore groups (Figure 54) and radically photopolymerized them under 900 nm wavelength light illumination. Furthermore, by employing the photocleavage capabilities of *o*-nitrobenzyl ether groups they spatially erased the microstructures at 700 nm [383].

Romano and co-workers applied direct laser writing techniques to inscribe micropatterns within thiol-ene photopolymers containing photolabile *o*-nitrobenzyl ester links without the requirement of a development step in the solvents [389]. The local change in the surface properties was further exploited to covalently attach bio-molecules.

Bialas et al. developed macromolecular dual photoresists based on the orthogonal reaction approach [364] by adding a photocaged diene generated from *o*-methyl benzaldehyde groups (oMBA) that was capable of undergoing dimerization under UV activation [390], as well as styrylpyrene units that formed [2π + 2π] dimers under visible light activation [391]. oMBA did not react under visible light illumination, while styrylpyrene dimerization was effectively avoided in the UV illumination spectrum, with maximum levels of photocaged oMBA reactivity (Figure 55) [364].

*O*-methyl benzaldehydes, *N*-ethyl-maleimides, and styrylpyrene groups have shown orthogonal reaction approaches with thermo-activated [4π + 2π] Diels–Alder chemistry [364].

Alves et al. utilized di-thioacetal-protected aldehyde groups in combination with oMBA to orthogonally induce deprotection of monomers along with the activation of oMBA by using visible light and UV light in a sequential manner. After activation, oMBA formed *o*-quinodimethane, which further underwent a thermally-induced [4π + 2π] cycloaddition reaction with suitable electron deficient dienophiles such as *N*-ethylmaleimide [392].

To summarize these aspects, the potential is great for combining state-of-the-art resins with photo-driven cycloaddition reactions. Such processes can proceed with excellent orthogonal chemistry and could offer various functional properties within a multi-material structure. Furthermore, broadening of the chemical library to include a greater range of mechanical and thermal properties could revolutionize the application areas, and a great market potential can hence be predicted for multi-materials.

From a hardware point of view, limited progress has been reported to date in dual-wavelength 3D printing. The number of commercially-available printers offering dual illumination projections are scarce, and those available suffer from low printing capacities. A strong need at present is to promote the manufacturing of 3D printers with greater manufacturing capacities (larger resin vats) along with the capability to modify the wavelength of light over a broad range. This could further expand the applicability of photoinitiators and monomers in multi-material 3D printing systems.

## 9. Conclusions

In summary, the progress in vat photopolymerization 3D printing, and especially multi-material fabrication, has made significant advancements in terms of the global efforts towards developing new materials which mimic nature. The applications of multi-material concepts in biological systems, stimuli responsive actuations, composites, self-healing, and functionally advanced materials in industry has been a center of attention over the past decade. However, as an emerging manufacturing process, there are certain challenges and bottlenecks in terms of material chemistry, printing capacity, hardware shortfalls, design strategies, purity, and the quality of fabricated structures.

Although additive manufacturing techniques based on extrusion and direct heat application have been advanced to a technically relevant level and are able to form 3D printed articles at low cost, they are certainly limited in terms of design complexity, surface finish, material resolution, raw materials portfolio, mechanical properties, and additional functionalities. Furthermore, the requirements of sintering and thermal processing require compatibility of different materials for processing together in order to form multi-material structures. The materials often lack appropriate interlayer adhesion (particularly at the layer interface of materials having different mechanical properties), which adversely affects multi-material fabrication.

Formation of multi-material structures via vat photopolymerization 3D printing provides several ways to improve interlayer adhesion, and is thus currently one of the most promising technologies for forming complex multi-material architectures with excellent resolution in a cost-effective manner. Multi-material vat photopolymerization processes have mostly been designed using multiple exchanging resin vats, which involves monotonous cleaning steps during resin changeover. Here, the efficiency of cleaning processes and trapped contaminations remains questionable. In contrast, grayscale vat photopolymerization 3D printing involves complex adjustments to the light intensity in a layer-by-layer approach in order to vary the crosslinking degree of the photopolymers in a controlled manner. However, this technology is limited by the number of reacting components, as the light is provided by a single light source. Orthogonal 3D printed polymer networks offer a promising solution to multi-material 3D-printing, however, the technology is in its infancy due to the limited commercial availability of functional monomers and 3D with the capability of changing the light source during the building of the object. Crossing such barriers could both revolutionize vat photopolymerization technology and bring material science into a new era of additive manufacturing where high-resolution multi-material structures can be fabricated with a high degree of design freedom and at a fast build speed.

With this detailed discussion of additive manufacturing, competitive technologies, vat photopolymerization, multi-material progress, and the underlying mechanisms and material chemistry, we are optimistic that this review provides readers with all of the key information for selection and implementation of effective multi-material strategies in general, and particularly in the field of photopolymerization.

In order to enhance the functionality, endurance, and integration of materials, the conjunction and streamlining of smart design, material chemistry, building software, and equipment is necessary, could ultimately stimulate the development of multi-materials, and could revolutionize the whole human community in coming years.

## Figures and Tables

**Figure 1 polymers-14-02449-f001:**
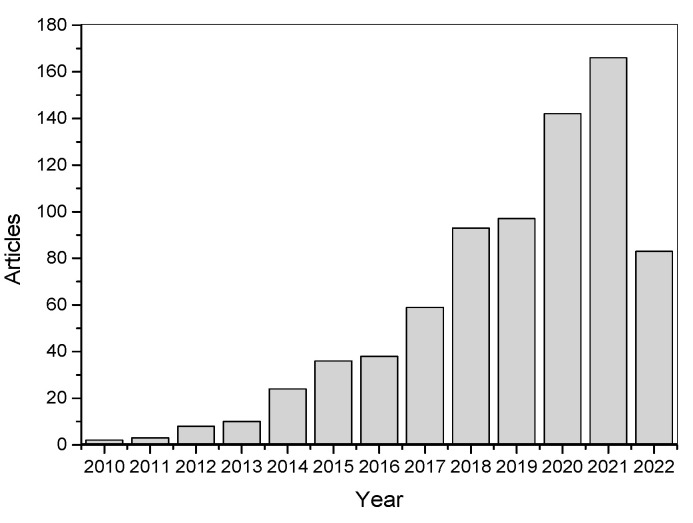
Number of research articles published on multi-material additive manufacturing after 2010 (source: Scopus).

**Figure 2 polymers-14-02449-f002:**
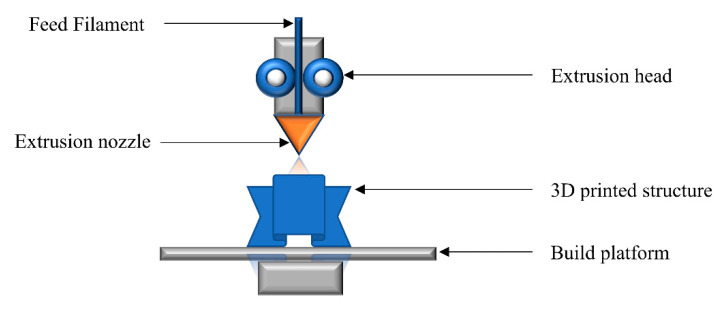
Schematic diagram of the fused filament fabrication (FFF) process. Redrawn from [10,93].

**Figure 3 polymers-14-02449-f003:**
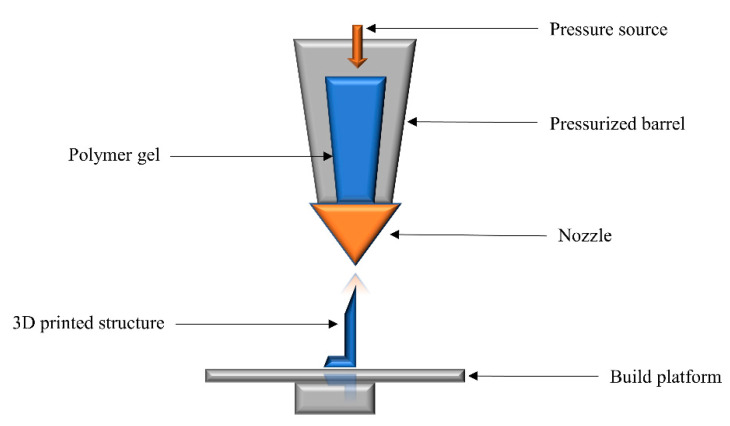
Schematic diagram of the direct ink writing (DIW) process. Redrawn from [10].

**Figure 4 polymers-14-02449-f004:**
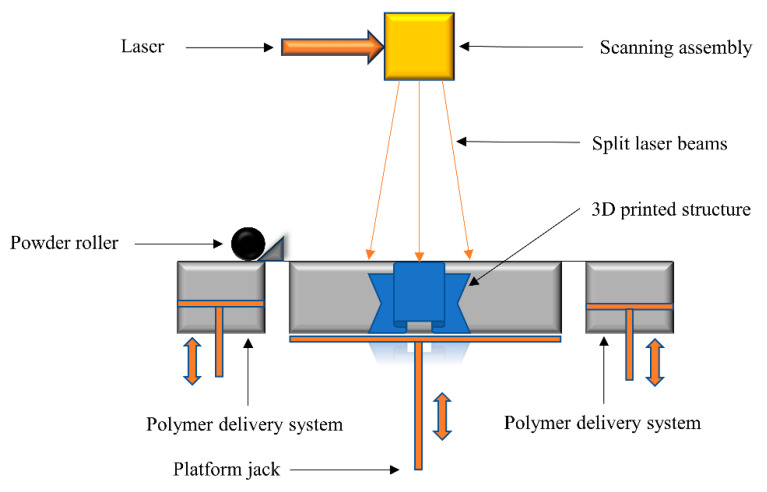
Schematic diagram of the selective laser sintering laser (SLS) process. Redrawn from [95].

**Figure 5 polymers-14-02449-f005:**
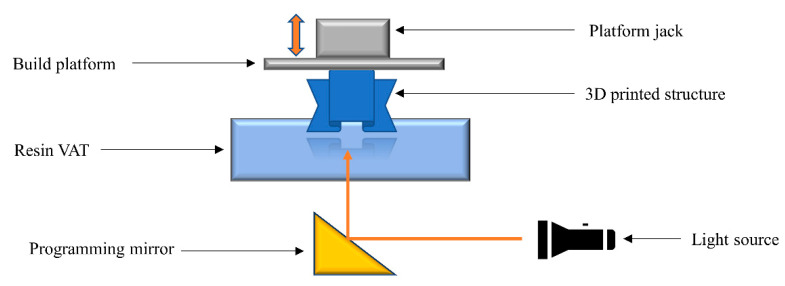
Schematic diagram of the stereolithography (SLA) process.

**Figure 6 polymers-14-02449-f006:**
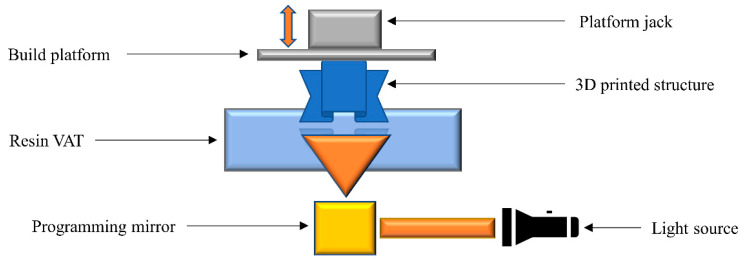
Schematic diagram of the digital light processing (DLP) process.

**Figure 7 polymers-14-02449-f007:**
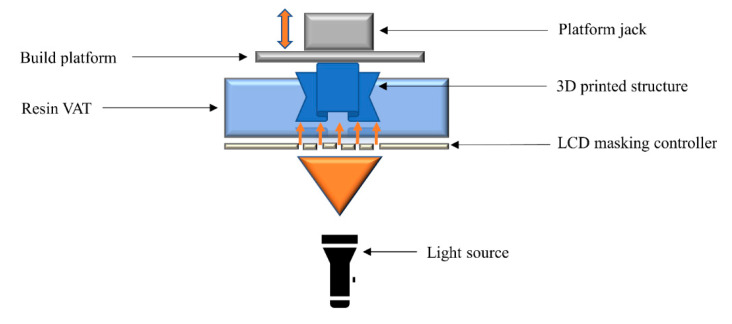
Schematic diagram of the liquid crystal display (LCD) printing process.

**Figure 8 polymers-14-02449-f008:**
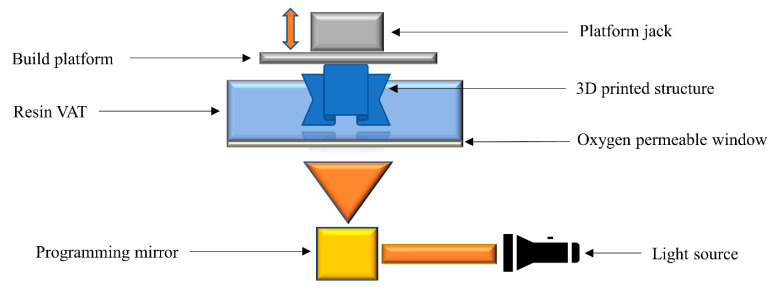
Schematic diagram of the continuous liquid interface printing (CLIP) process. Redrawn from [106].

**Figure 9 polymers-14-02449-f009:**
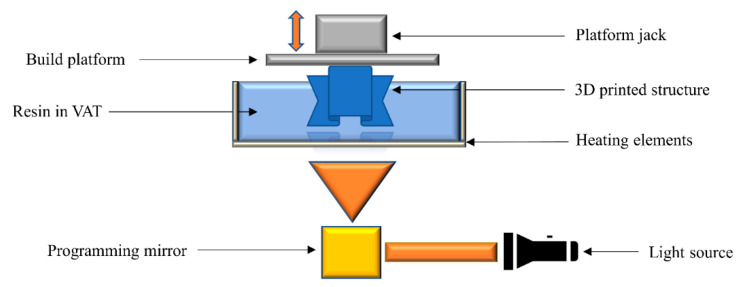
Schematic diagram of the hot lithography process.

**Figure 10 polymers-14-02449-f010:**
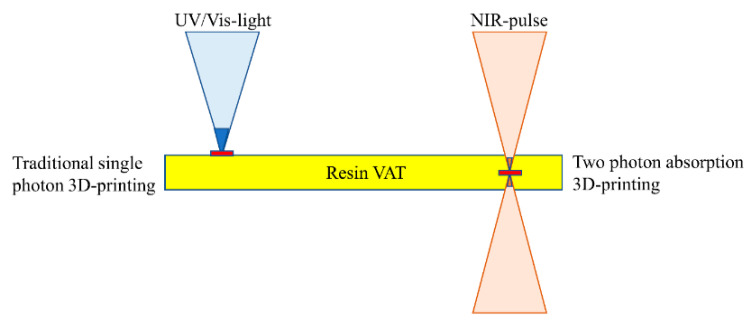
UV light is absorbed at the surface of a photosensitive resin, and can only be used for fabrication of planar structures (one-photon absorption). NIR light is focused into the volume of the photocurable resin and is used for 3D structuring (TPP). Reproduced with permission from [123]. Copyright © 2022 Elsevier.

**Figure 11 polymers-14-02449-f011:**
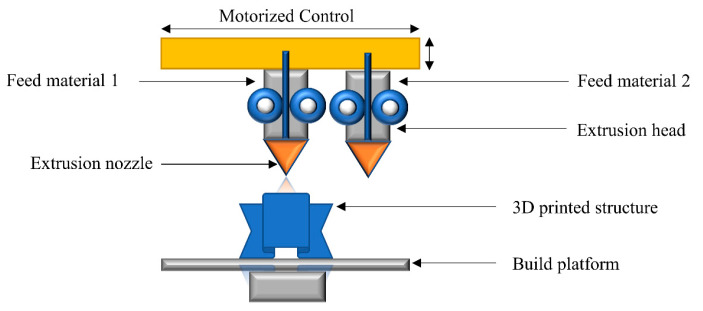
Schematic diagram of the multi-material FFF process. A central motorized controller adjusts the position of the extrusion heads intermittently for layer-by-layer extrusion of each material.

**Figure 12 polymers-14-02449-f012:**
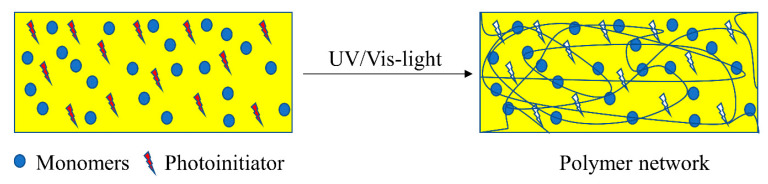
Schematic representation of a photo-curable resin prior to and after light exposure, high-lighting the formation of an insoluble and infusible polymer network.

**Figure 13 polymers-14-02449-f013:**
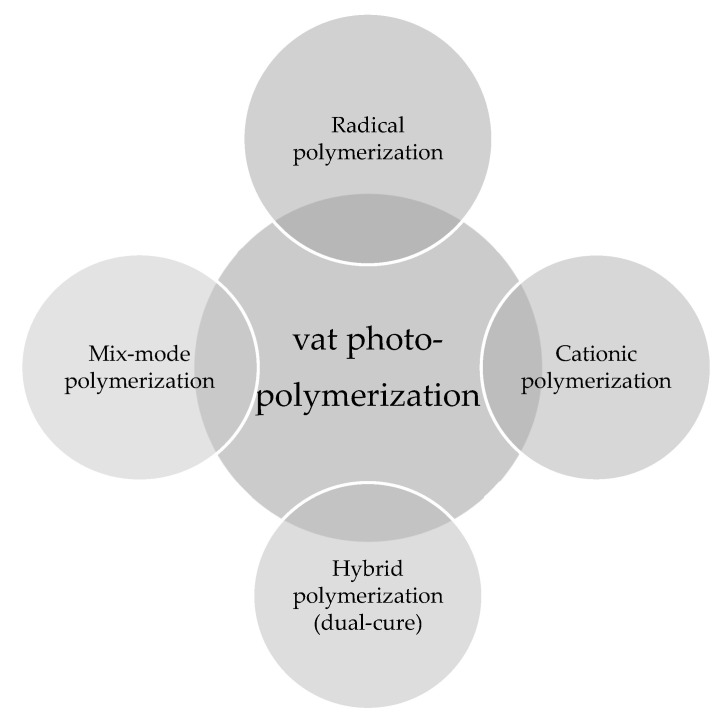
Reaction strategies for vat photopolymerization 3D printing.

**Figure 14 polymers-14-02449-f014:**
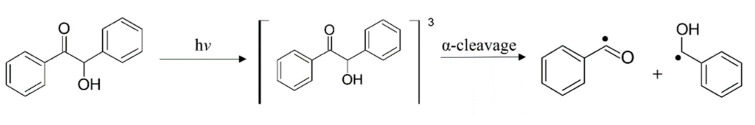
Type I photoinitiation mechanism [164]. Copyright © 2022 ACS.

**Figure 15 polymers-14-02449-f015:**
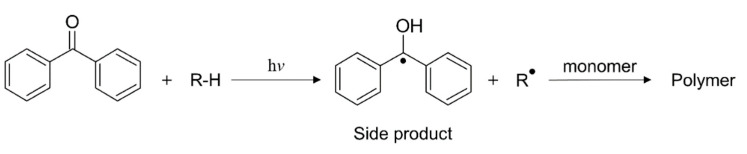
Type II photoinitiation mechanism [164]. Copyright © 2022 ACS.

**Figure 16 polymers-14-02449-f016:**
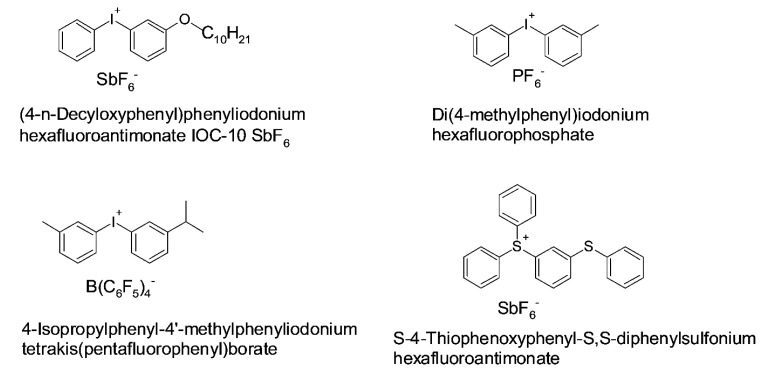
Structure of selected cationic photoinitiators. Reproduced with permission from [186]. Copyright © 2006 John Wiley and Sons.

**Figure 18 polymers-14-02449-f018:**
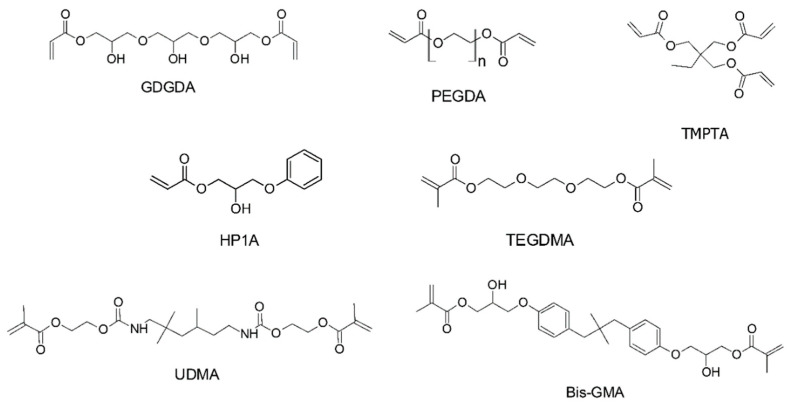
Structures of selected acrylates and methacrylate monomers typically used for vat photopolymerization-based 3D printing.

**Figure 19 polymers-14-02449-f019:**
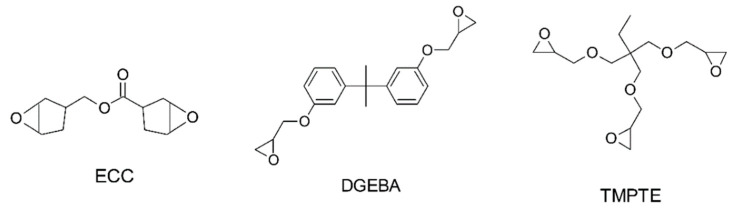
Structure of epoxy monomers commonly used for vat photopolymerization 3D printing.

**Figure 20 polymers-14-02449-f020:**
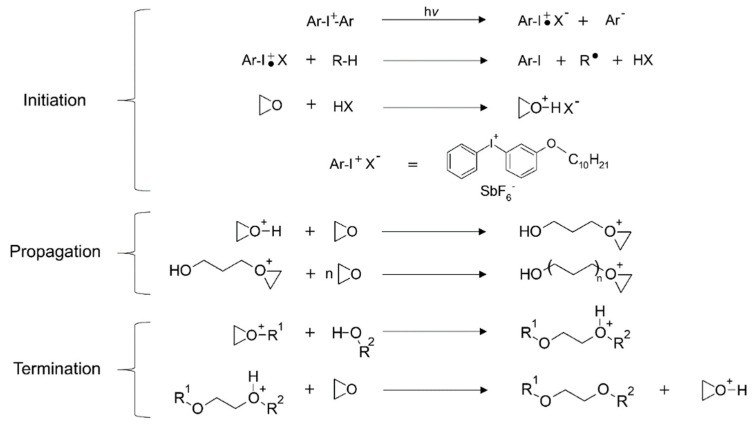
Mechanism of photoacid-initiated epoxy polymerization (cationic chain-growth). Reproduced with permission from [219], Copyright © 2022 John Wiley and Sons.

**Figure 21 polymers-14-02449-f021:**
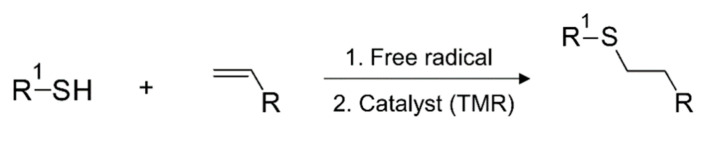
General reaction scheme of the addition of thiols across “ene” functions: radical mediated thiol-ene versus catalyzed thiol-Michael reaction (TMR) [225].

**Figure 22 polymers-14-02449-f022:**
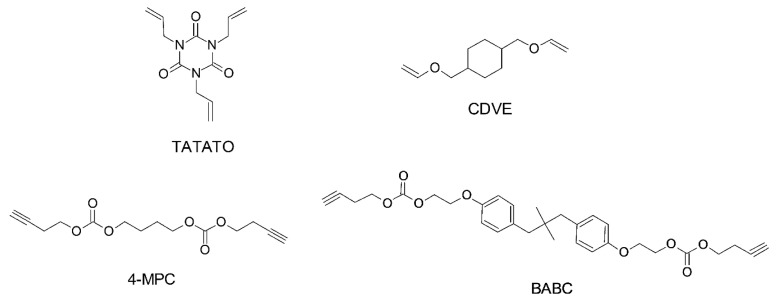
Structure of selected alkenes and alkynes used for vat photopolymerization 3D printing.

**Figure 23 polymers-14-02449-f023:**
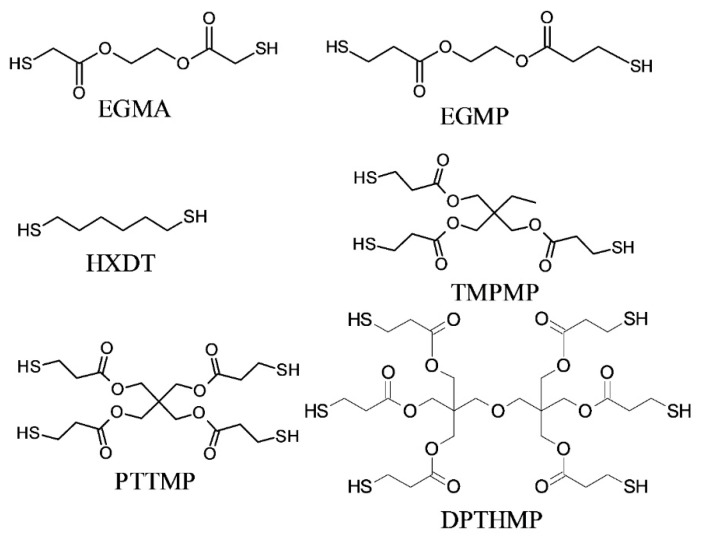
Structures of selected commercially available thiols.

**Figure 24 polymers-14-02449-f024:**
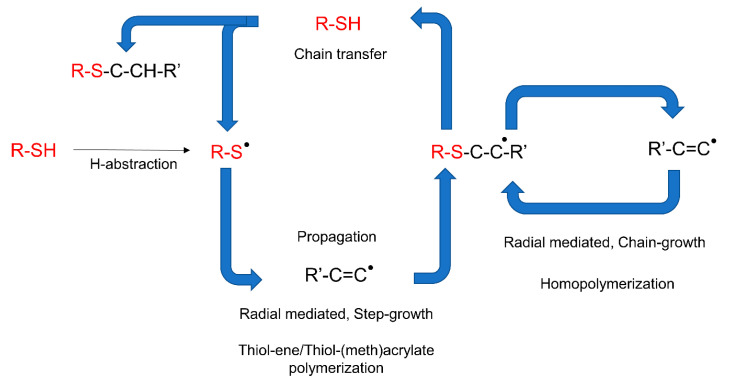
Reaction cycle representing mixed-mode thiol-ene and acrylate homopolymerization. Active carbon centers can participate in chain transfer as well as chain propagation. Reproduced with permission from [225]. Copyright © 2022 John Wiley and Sons.

**Figure 25 polymers-14-02449-f025:**
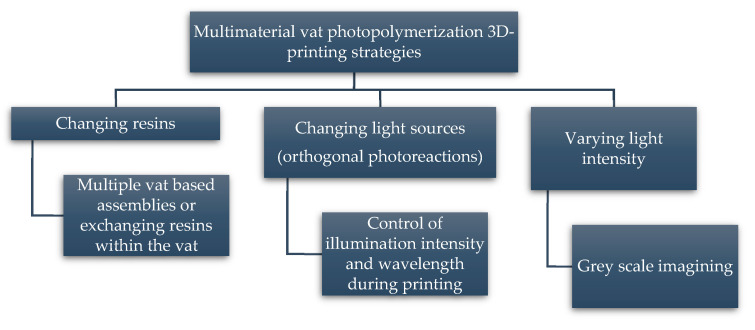
Strategies for multi-material vat photopolymerization 3D printing.

**Figure 26 polymers-14-02449-f026:**
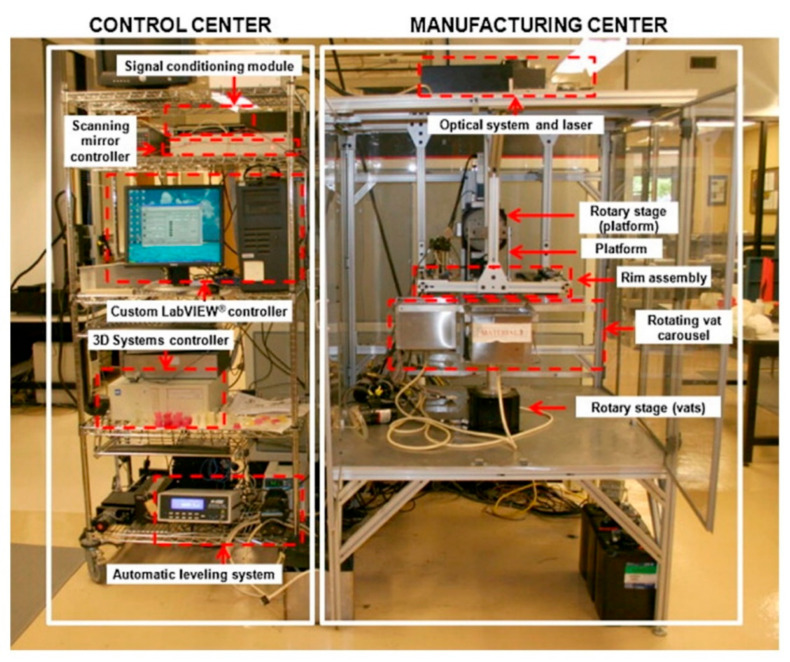
Multi-material SLA setup developed by Choi and coworkers, reproduced with permission from [57]. Copyright © 2022 Elsevier.

**Figure 27 polymers-14-02449-f027:**
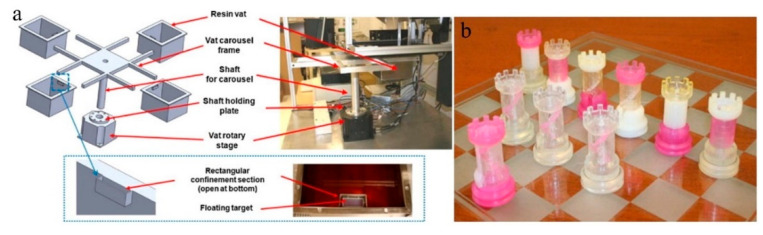
(**a**) Multi-vat carousel assembly employed in SLA 3D printing and (**b**) multi-material 3D printed articles using SLA. Reproduced with permission from [57]. Copyright © 2022 Elsevier.

**Figure 28 polymers-14-02449-f028:**
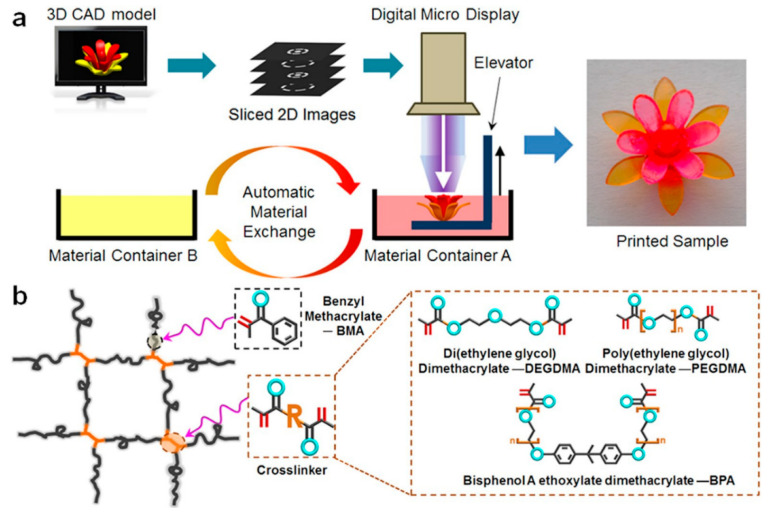
(**a**) Process working schematics for multi-material SLA printing using two different monomers and (**b**) the chemical structure developed by monomers via multi-material SLA printing. Reproduced from [61].

**Figure 29 polymers-14-02449-f029:**
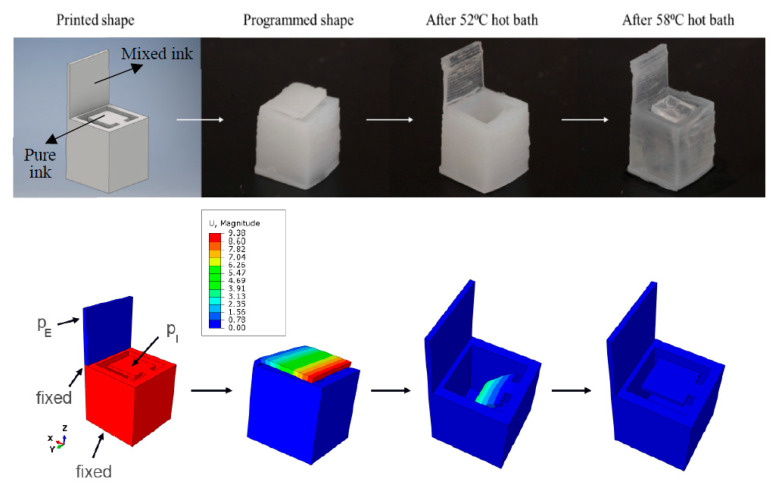
Thermally-induced transitions in a double-lid multi-material DLP printed box. Both lids remain closed below 52 °C; at 52 °C, both lids open. The inner lid closes at 58 °C. Reproduced from [70].

**Figure 30 polymers-14-02449-f030:**
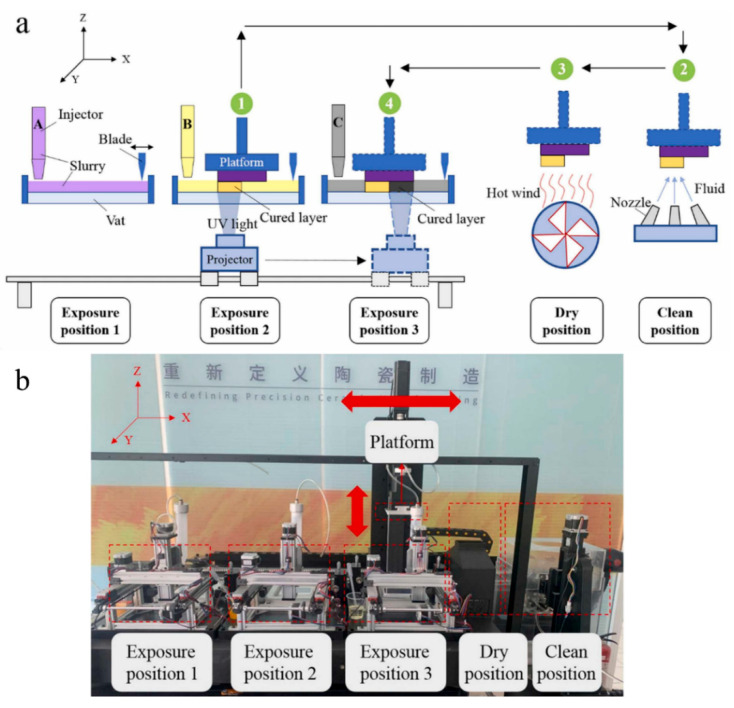
(**a**) Conceptual diagram of multi-material DLP 3D printing setup and (**b**) overview of the actual multi-material DLP 3D printer developed by Hu and co-workers. Reproduced with permission from [63]. Copyright © 2022 Elsevier.

**Figure 31 polymers-14-02449-f031:**
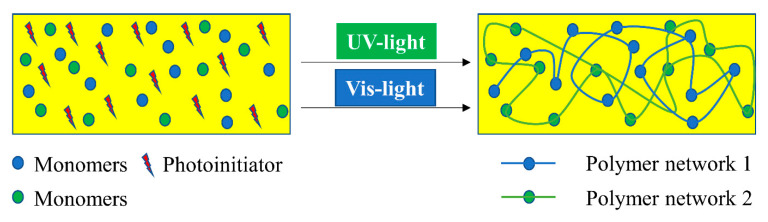
Conceptual illustration of orthogonally-evolved photopolymer networks via two different types of light illumination. Polymer Network 1 forms the first layer and polymer Network 2 forms the second layer for subsequent building of multi-material structures.

**Figure 32 polymers-14-02449-f032:**
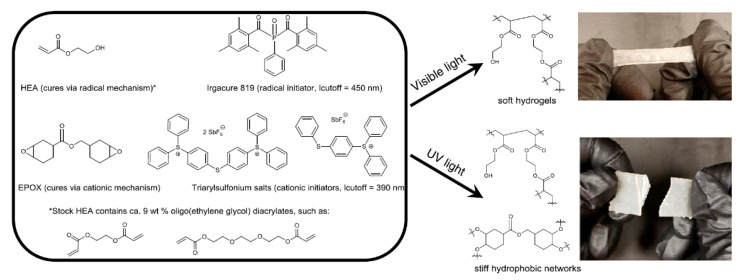
Representation of MASC strategy to achieve different networks with discrete wavelengths of light irradiation. A stiffer network was formed by cationic epoxy polymerization, while a softer network was generated by radical acrylate polymerization. Reproduced from [320].

**Figure 33 polymers-14-02449-f033:**
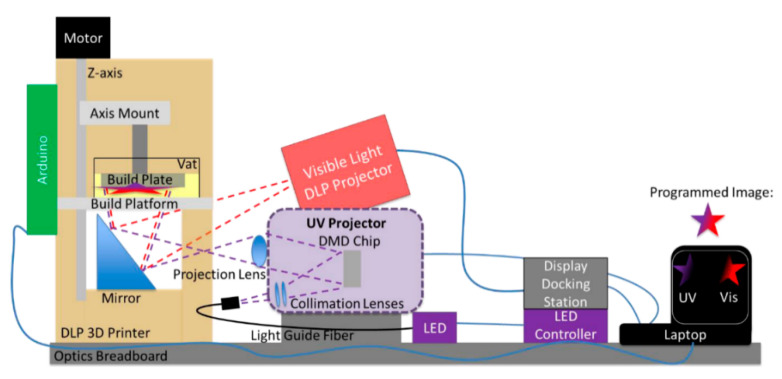
Customized multi-material DLP 3D printing setup with visible and UV projectors. Scanning process control was carried out using an integrated laptop. Reproduced from [320].

**Figure 34 polymers-14-02449-f034:**
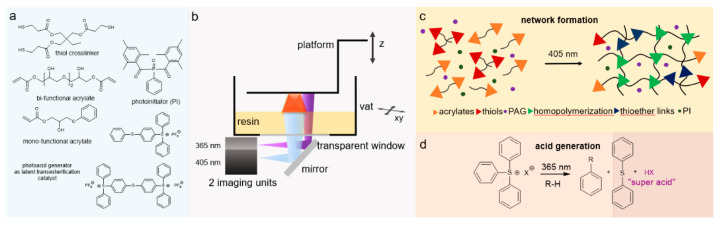
(**a**) Constituents of the resin formulation for selectively activating a transesterification catalyst in thiol-acrylate photopolymers; (**b**) dual-light wavelength DLP 3D printer setup; (**c**) visible light-mediated network formation during DLP 3D printing; (**d**) UV light-triggered activation of Brønsted acids acting as a transesterification catalyst. Reproduced from Ref. [237] with permission from the Royal Society of Chemistry.

**Figure 35 polymers-14-02449-f035:**
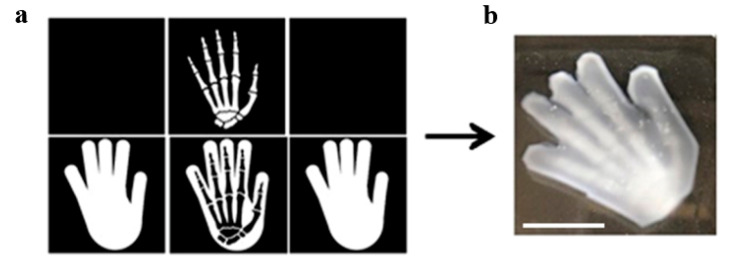
(**a**) Arbitrary scale DLP projection images; upper design (bones) printed with UV light, lower design printed with visible-light. (**b**) Multi-material DLP 3D printed hand using UV and visible light projections. Scale bar represents 25 mm. Reproduced from [320].

**Figure 36 polymers-14-02449-f036:**
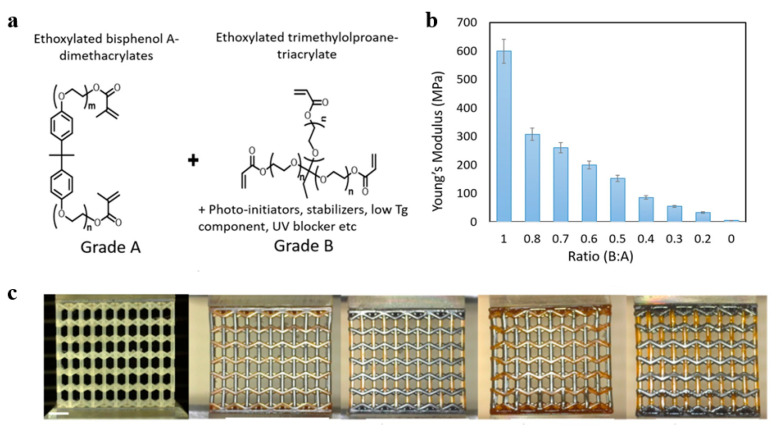
(**a**) Components of resin formulation; (**b**) tunable stiffness bandwidth in photocured polymers by varying concentration of A and B components; and (**c**) physical representation of printed architectures with varying compositions. Reproduced from [327].

**Figure 37 polymers-14-02449-f037:**
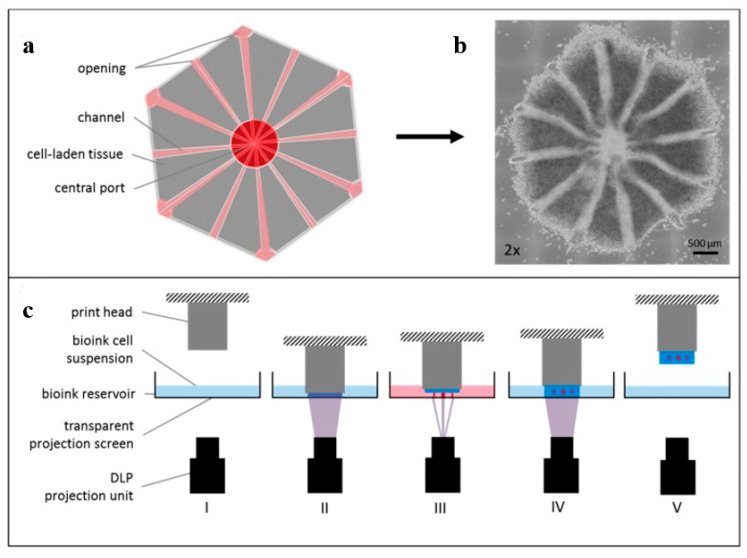
Cell brick 3D printing process of liver: (**a**) virtual 3D design; (**b**) microscopic image of the printed multi-material cell-laden hydrogel; (**c**) multi-material DLP 3D printing setup used for liver design printing. Reproduced from [329].

**Figure 38 polymers-14-02449-f038:**
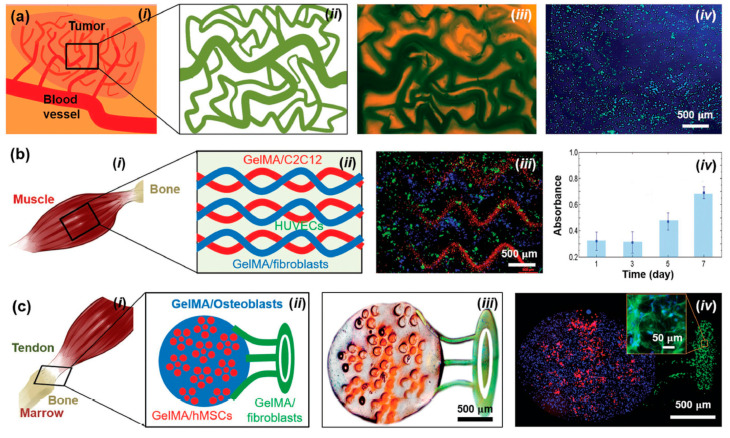
(**a**) Tumor angiogenesis model: (**i**) illustration of tumor angiogenesis model; (**ii**) schematic of the mask for 3D printing; (**iii**) bioprinted MCF7 cell-laden microvascular bed of GelMA (blue); (**iv**) bioprinted cell-laden microvascular bed of GelMA (blue) seeded with cells (green) in the channels. (**b**) Skeletal muscle model: (**i**) schematic showing the skeletal muscle tissue; (**ii**) schematic of the mask for 3D printing; (**iii**) bioprinted structure of GelMA containing patterned cells (red) and fibroblasts (blue); (**iv**) Presto blue measurements of cell proliferation in the printed structures. (**c**) Tendon-to-bone insertion model: (**i**) schematic of the tendon-to-bone insertion site; (**ii**) schematic of the mask for printing; (**iii**) bright field optical image showing a bioprinted dye-laden GelMA structure; (**iv**) bioprinted structure of GelMA containing patterned osteoblasts (blue), MSCs (red), and fibroblasts (green). Produced with permission from [331]. Copyright © 2022 John Wiley and Sons.

**Figure 39 polymers-14-02449-f039:**
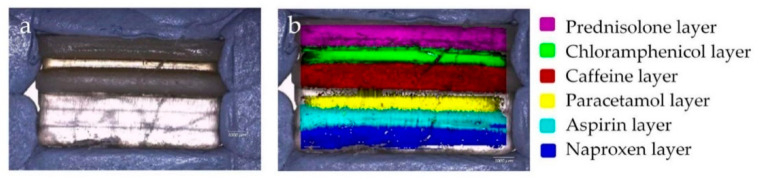
Visual presentation of a (multilayer) polypill under: (**a**) optical microscope and (**b**) Raman mapping. Each layer presents a different drug printed layer. Work reproduced from [332].

**Figure 40 polymers-14-02449-f040:**
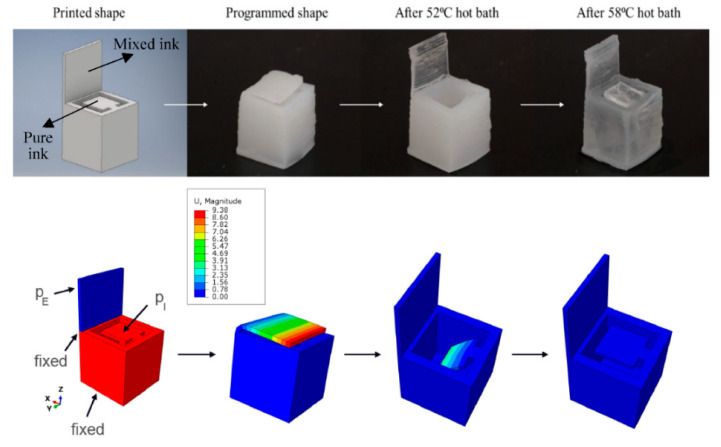
(**Top**) 3D printed box with two lids. The box opens at 52 °C and closes at 58 °C. (**Bottom**) Results of the finite element simulation. Red and blue colors in the first figure on the left denote pure and mixed ink, respectively. Reproduced from [70].

**Figure 41 polymers-14-02449-f041:**
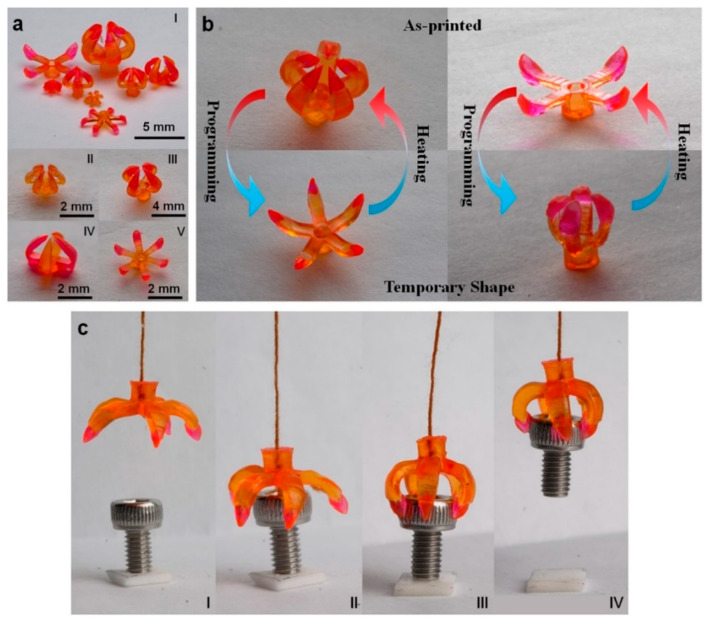
3D printed multi-material grippers: (**a**) different sizes and designs of 3D printed grippers; (**b**) illustration of transitions between printed and temporary shapes as a function of temperature; (**c**) gripper used to lift a bolt assembly. Reproduced with permission from [61].

**Figure 42 polymers-14-02449-f042:**
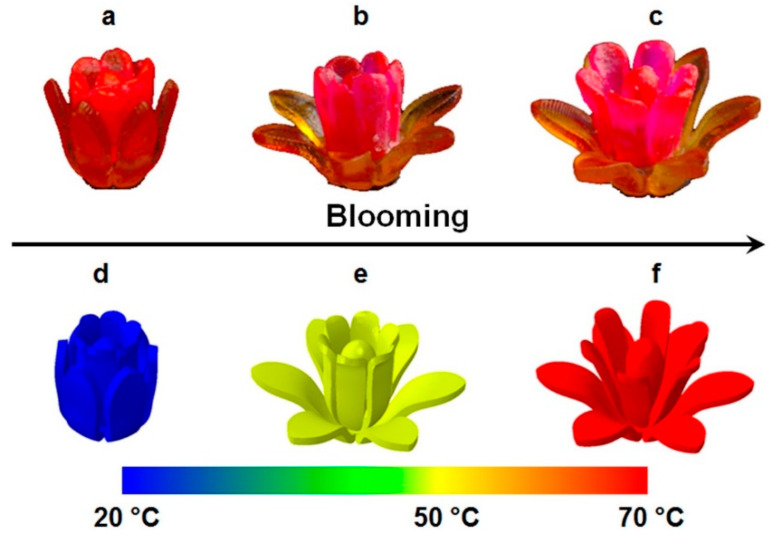
Stimuli-responsive behavior of a multi-material 3D printed flower: (**a**) Programmed (temporary) shape of the multi-material flower held at 20 °C; (**b**) heating to 50 °C, resulting in the flower’s outer buds opening; (**c**) the flower fully bloomed at 70 °C; (**d**–**f**) simulation images of the complete multi-material flower blooming process. Reproduced with permission from [61].

**Figure 43 polymers-14-02449-f043:**
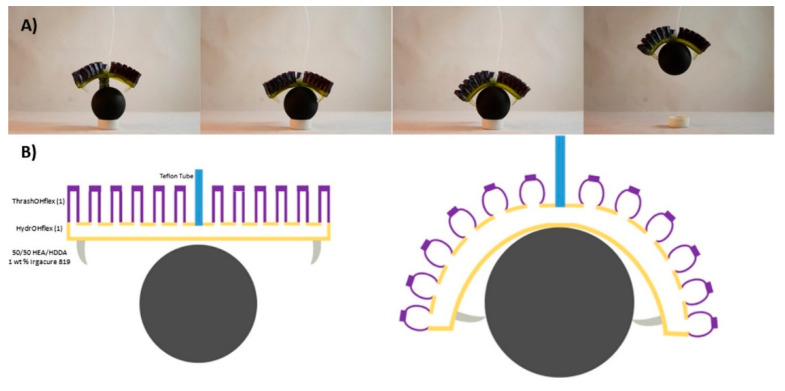
(**A**) Multi-material 3D printed pneumatic gripper, descending, pneumatically actuating, and grabbing a plastic ball. (**B**) Schematic illustration of the multi-material gripper before (**left**) and after (**right**) pneumatic actuation. Ball diameter = 38 mm. Reproduced from [347].

**Figure 44 polymers-14-02449-f044:**
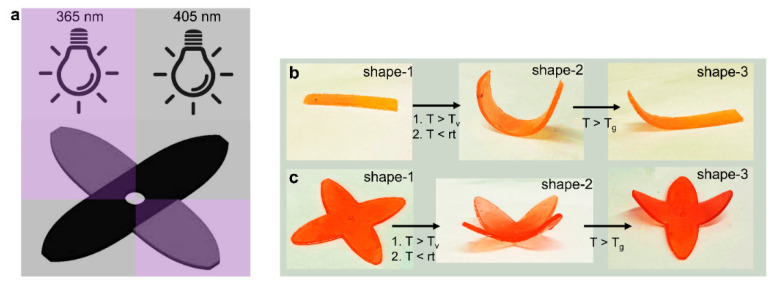
(**a**) Illustrations of the printing setup of the rectangular bar and grippers; (**b**) the permanent, programming, and spatially-controlled shaping experiment; and (**c**) the permanent, programming, and spatially-controlled state of the multi-material gripper under thermal stimuli. Reproduced from Ref. [237] with permission from the Royal Society of Chemistry.

**Figure 45 polymers-14-02449-f045:**
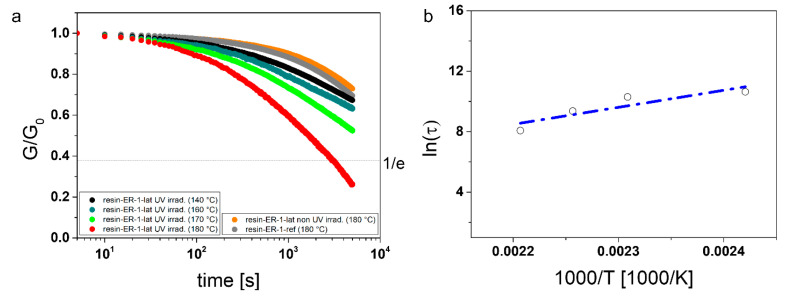
(**a**) Normalized stress relaxation behavior of multi-materials over time at various temperatures; (**b**) Arrhenius plot of UV-activated multi-materials derived from measured relaxation times. Reproduced from Ref. [237] with permission from the Royal Society of Chemistry.

**Figure 46 polymers-14-02449-f046:**
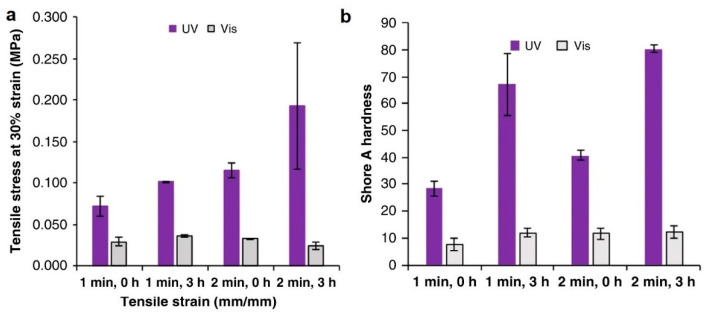
(**a**) Tensile stress values at 30% strain for UV and vis-light cured samples; (**b**) Shore A hardness of UV and vis-light cured samples. Time in minutes represents illumination times and time in hours represents times of thermal post-curing [320].

**Figure 47 polymers-14-02449-f047:**
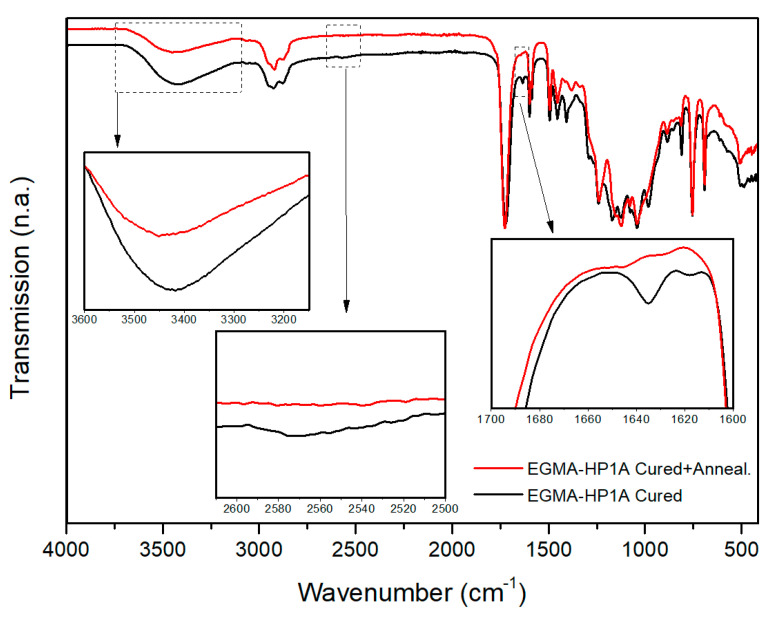
ATR-FTIR spectra of thiol-acrylate 3D printed samples prior to and after thermal annealing at 180 °C for 4 h. Inset plots present zoomed images of the -OH (3600–3200 cm^−1^), -SH (2600–2500 cm^−1^), and C=C-H (1700–1600 cm^−1^) absorption bands before and after thermal annealing. Reproduced with permission from [101]. Copyright © 2022 Elsevier.

**Figure 48 polymers-14-02449-f048:**
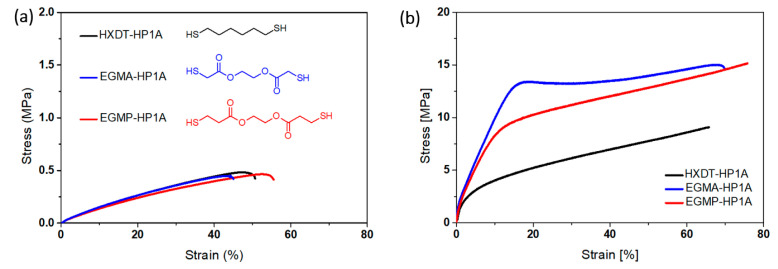
Mechanical properties of DLP 3D printed materials (**a**) prior to and (**b**) after thermal treatment at 180 °C for 4 h. Reproduced with permission from [101]. Copyright © 2022 Elsevier.

**Figure 49 polymers-14-02449-f049:**
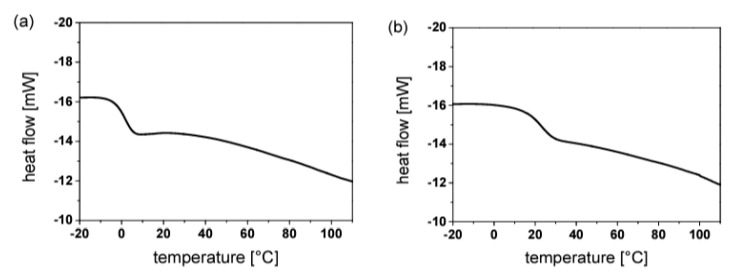
DSC plots of vat photopolymerized thiol-acrylate resins (**a**) prior to and (**b**) after thermal post-treatment at 180 °C for 4 h. Reproduced from Ref. [102] with permission from the Royal Society of Chemistry.

**Figure 50 polymers-14-02449-f050:**
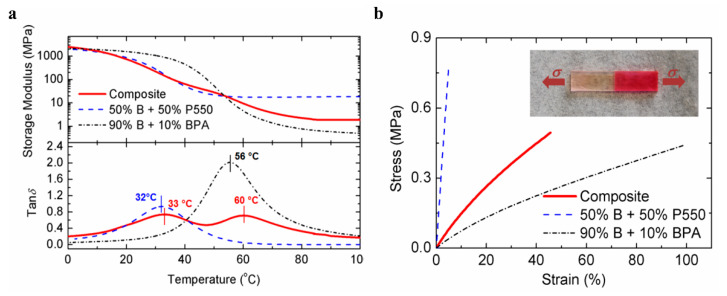
Thermomechanical tests to investigate the interface bonding within multi-material structures (Material A = 50%B + 50%P550; Material B = 90%B + 10%BPA): (**a**) DMA and (**b**) uniaxial tensile tests of the composite and the single materials [61].

**Figure 51 polymers-14-02449-f051:**
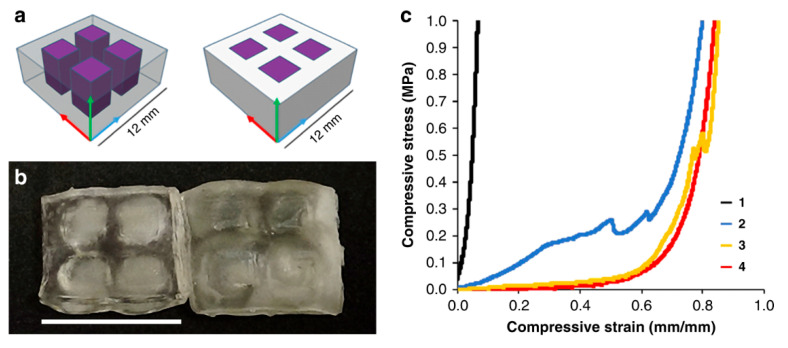
(**a**) 3D model of four-pillar multi-material objects. Purple corresponds to pillars printed with UV light, while the white/transparent outer region corresponds to domains printed with visible light. (**b**) Printed multi-material structures prior to thermal post-curing (left) and after thermal post-curing at 60 °C for 3 h (right). (**c**) Representative compressive stress–strain plots of multi-material 3D printed test specimens. Black curve (1) = Sample cured with UV light. Blue curve (2) = Multi-material sample compressed along the z-axis. Yellow curve (3) = Multi-material sample compressed along the x-axis. Red curve (4) = Sample cured with visible light [320].

**Figure 52 polymers-14-02449-f052:**
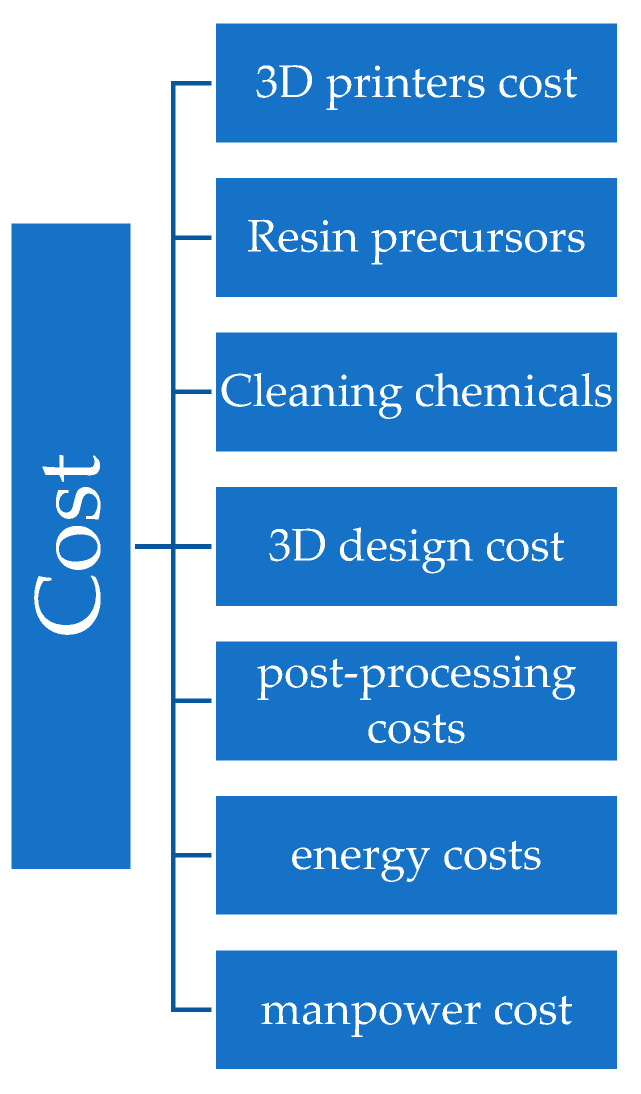
Investment and running costs in vat photopolymerization 3D printing technology.

**Figure 53 polymers-14-02449-f053:**
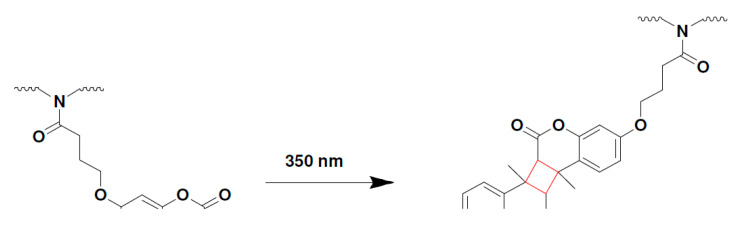
Coumarin side-groups undergoing [2π + 2π] cycloaddition reaction upon UV irradiation. Reproduced with permission from [375]. Copyright © 2022 Elsevier.

**Figure 54 polymers-14-02449-f054:**
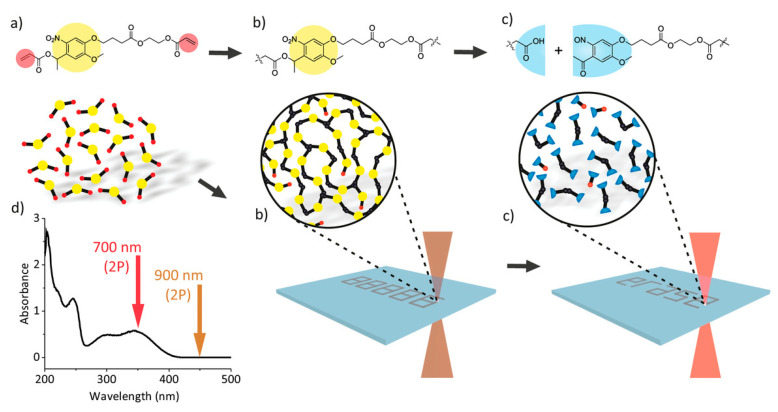
Representational diagram of printing and cleaving process: (**a**) structure of acrylate with pendant *o*-nitrobenzyl ester (ONB) monomer; (**b**) micro-fabrication using DLW (900 nm wavelength) via radical photo-polymerization of acrylates; (**c**) selective cleavage of the resist at 700 nm wavelength through *o*-NB cleavage; (**d**) UV-vis spectrum of the ONB monomers. Arrows indicate the corresponding two-photon wavelengths for printing and cleaving. Reproduced with permission from [383]. Copyright © 2022 John Wiley and Sons.

**Figure 55 polymers-14-02449-f055:**
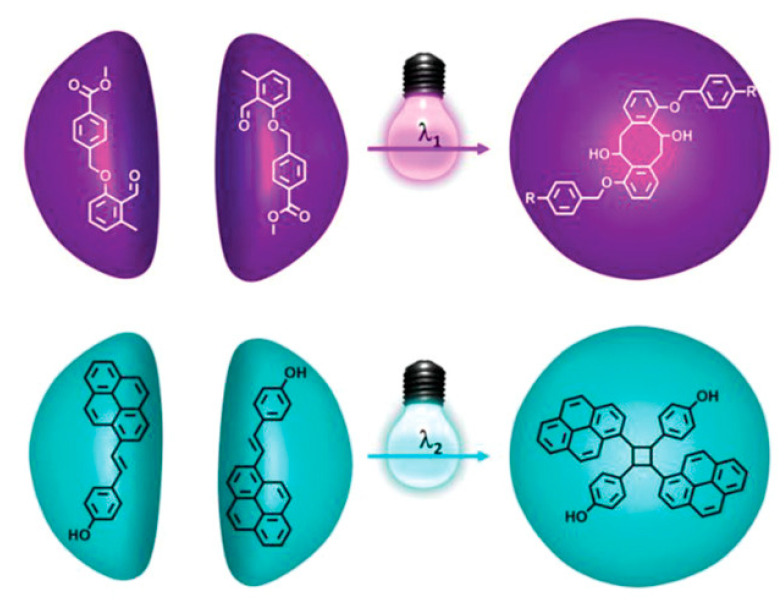
Purple area: o-MBA (λ_1_ maximum = 330 nm purple); turquoise area: StyP (λ_2_ maximum = 435 nm): activations for implementing a λ-orthogonal photoresist. Reproduced with permission from [364]. Copyright © 2022 John Wiley and Sons.

**Table 1 polymers-14-02449-t001:** Young’s moduli of selected soft and rigid materials [1,14,22].

Category	Material	Young’s Modulus (Pa)
Soft	Brain tissues	10^3^
	Alginate hydrogel fat	10^4^
	Silicone elastomer	10^5^
	Polydimethylsiloxane	0.8 × 10^6^
	Polycaprolactone	0.2 × 10^7^
	Biological skin	0.5 × 10^8^
	Rubber	0.8 × 10^8^
	Polyethylene (low density)	0.5 × 10^9^
Rigid	Polylactic acid	0.5 × 10^10^
	Nylon	0.6 × 10^10^
	Wood	10^10^
	Polymethylmethacrylate	4.5 × 10^10^
	Polyethylene terephthalate	5.5 × 10^10^
	Bone	0.5 × 10^11^
	Glass	0.8 × 10^11^
	Copper	10^11^
	Steel	0.5 × 10^12^
	Diamond	10^12^

**Table 2 polymers-14-02449-t002:** Different printing strategies and applications of TPA polymerizations.

	Processing Strategy	Application Area	Resolution Achieved	Reference
TPA Photo polymerization	Radical quencher	Microstructures	100 nm	[124]
	Activation beam	Microdevices	40 nm	[125]
	Scanning speed manipulation	Micromachines	25 nm	[126]
	Self-smoothing	Micro-optics	20 nm	[127]

**Table 3 polymers-14-02449-t003:** Characteristics of selected common 3D printing technologies.

	Suitable Materials	Resolution	Building Speed	Benefits	Limitations	Reference
Fused filament fabrication (FFF)	Acrylonitrile butadiene styrene, nylon, polylactic acid, polyethylene terephthalate, polyvinylalcohol, high impact polystyrene, thermoplastic polyurethane, polycarbonate, polypropylene	100 µm	1–10 m/min	Ability to print large functional materials with standard thermoplastic polymers	Low resolution, nozzle choking, non-homogenous filament melting, material, anisotropy along z-axis	[1,62,68,75,132,133]
Direct ink writing (DIW)	Liquid polymer/melt/gel/paste	1–100 µm	3 m/min	Highest resolution among all extrusion processes	Higher cost, not suitable for complex geometries	[1,62,68,134,135,136]
Selective laser sintering (SLS)	Polylaurylamide, polyether ketone ketone, polyamide, polycaprolactone	80–100 µm	1.80–3.6 m/min	Ability to process standard plastics, good mechanical properties	Lower resolution, rough surface	[1,10,68,137,138,139]
Stereolithography (SLA)	Acrylate, methacrylate, epoxy, vinyl monomers	5–50 µm	0.25 mm/min	High resolution and accuracy	Limited availability of photopolymers, toxicity of monomers	[1,68,99,138,140,141,142,143,144]
Digital light processing (DLP)	Acrylate, methacrylate, epoxy, vinyl monomers	5–50 µm	0.4–2.5 mm/min	High resolution and accuracy, lower cost, higher printing speed compared to SLA	Limited availability of photopolymers, toxicity of monomers	[1,68,143]
Liquid crystalline display (LCD)	Acrylate, methacrylate, epoxy, vinyl monomers	<50 µm	10 mm/min	High resolution and accuracy, lower cost	Limited availability of photopolymers, toxicity of monomers	[37,103,104,105]
Continuous liquid interface printing (CLIP)	Acrylate, methacrylate, vinyl monomers, epoxies	<100 µm	8–16 mm/min	High printing speed	Anisotropy of printed structures	[106,109]
Two-photon absorption (TPA)	Acrylate, methacrylate, vinyl monomers, epoxies	<100 nm	0.08–33 mm^3^/min	Excellent resolution	Expensive, time consuming, requires tedious control strategies (rastering)	[145,146]
Volumetric 3D printing	Acrylate, methacrylate, vinyl monomers, epoxies	Up to 80 µm	10 mm/min	Fast printing speed.	High viscosity resin (>10 Pa·s) required. Costly technology, tedious resin formulation strategies, low absorption and high reactivity of monomers required.	[86,128,130,147]
Solution mask liquid lithography (SMaLL)	Acrylate, methacrylate, vinyl monomers, epoxies	Up to 100 µm	8.33 mm/min	Large curing depth, no moving parts required, rapid curing rates	Additional photochromes and sensitizer required. Reaction strategies must be developed before printing.	[24]

**Table 5 polymers-14-02449-t005:** Multi-material vat photopolymerization.

Technology	Strategy	Reference
SLA	Carousel-like rotating disks including various resins	[297]
SLA	Carousel-based rotary vats	[298]
SLA	Multiple resin injections under the servo-stage on the building stage through an orifice	[299]
SLA	Resin droplet delivery via rotary wheel	[300]
DLP	Resin exchange in vat with intermittent cleaning	[301]
µSLA	Multiple-resin dynamic liquid control within an integrated fluidic cell. Pumps for drawing/withdrawing of materials.	[302]
SLA	Multiple resin supply via microchannels	[303,304]
SLA	Multiple resin injection and intermittent cleaning	[305]

**Table 6 polymers-14-02449-t006:** Applications of multi-material vat photopolymerization in various bio-medical areas.

Applications	Printing Materials	Methodology	Reference
Scaffolds for tissue engineering	PEGDA, PEGDMA	Resin exchange, SLA	[334]
Multilayer polypills	PEGDA + dissolved drug	Resin exchange, SLA	[332]
Multilayer polypills	PEGDA, PEGDMA + dissolved drug	Resin exchange, SLA	[333]
Tissue-porous scaffolds	PEGDA, Commercial photocurable resins, + Leachable salt particulates	DLP	[335]
Bioactive scaffolds	PEGDA, PEGDMA, + fluorescently labeled components	Resin exchange, SLA	[336]
Neovasculature	PEGDA, PEGDMA, + Murine cells	Resin exchange, SLA	[337]
Constructs with Encapsulated Cells	PEGDA + human dermal fibroblasts	Resin exchange, SLA	[338]
Piezoelectric acoustic sensor	PEGDA + barium titanate nanopowder + multi-walled carbon nanotubes	Resin exchange, DLP	[339]
Cell encapsulation	PEGDA + cells	Resin exchange, SLA	[340]
Multi-material cantilevers	poly(ethylene glycol) diacrylate (PEGDA) and acrylic-PEG-collagen (PC)	Resin exchange, SLA	[341]
Selective Porous Barriers	PEGDA: MW 258, MW 575, MW 700	Resin exchange, SLA	[65]
Biological sensors	PEGDA, commercial resin, + biomolecules	Resin exchange, SLA	[342]
Tissue scaffolds	PEGDA+ fluorescently-labeled polystyrene microparticles	Resin exchange, SLA	[343]
Spatially-designed biological sensor	oxidized methacrylic alginate, poly(ethylene glycol) methyl ether methacrylate, PEGDA + cells	Resin exchange, SLA	[344]
Cells interactive sensors	Gelatin methacrylate, PEGDA, fluorescent dextrans, cells	Resin exchange, SLA	[345]

**Table 7 polymers-14-02449-t007:** Stimuli-responsive materials printed by multi-material vat photopolymerization.

Applications	Stimuli	Printing Materials	Multimaterial Strategy	Reference
Hydrogel cantilevers and actuators	Chemical stimuli	Poly(ethylene glycol) diacrylate (PEGDA) and acrylic-PEG-collagen (PC) mixtures	Resin exchange in vat, SLA	[341]
Hydrogels	Thermal stimuli	Poly(*N*-isopropylacrylamide), *N*,*N*′-Methylene-bis(acrylamide) mixtures	Resin exchange in vat, SLA	[348]
Hinges, robotic arms, bars, and sheets	Thermal stimuli	Bisphenol A ethoxylate diacrylate (BPADA), glycidyl methacrylate (GMA), *n*-butyl acrylate (BA), a diamine cross-linker [poly(propylene glycol) bis(2-aminopropyl ether); D230], mixtures	Grayscale DLP 3D printing	[321]
Hydrogels	Osmotic pressure, temperature and pH	*N*-Isopropylacrylamide, 2-carboxyethylacrylate, *N*,*N*′-ethylenebisacrylamide	1. Swelling rates via high surface area patterning, 2. Crosslinking density via photo-exposure, 3. Chemical composition via resin vat exchange	[349]

**Table 8 polymers-14-02449-t008:** Tensile properties of single and multi-material structures [61].

	Material A	Material B	Multi-Material
Modulus (MPa)	16.5	0.92	1.84
Failure strain	5%	99%	46%

## Data Availability

Not applicable.

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
