# Peer review of "A Review of Multi-Material 3D Printing of Functional Materials via Vat Photopolymerization"

_polymers, 2022, doi:10.3390/polym14122449_

Round 1

Reviewer 1 Report

Manuscript with number “polymers-1752213” was reviewed:

The introduction needs some improvements. Use more recent references for writing this section.

There is a lake of information about the following topics:

Economical study of the process

              Post-processing

              Usual defects

              Thermal analysis of process

Please compare this process with other methods such as MEX.

Please add some information about the mechanical properties of the sample produced by this method.

The conclusion needs more work, please add some information about the advantages and limitations of this process compared with other polymer 3D printing techniques.

Following papers have been suggested for introduction and comparing results:

Experimental investigations into abrasive flow machining (AFM) of 3D printed ABS and PLA parts

Temperature-compensated constitutive model of fused filament fabrication 3D printed PLA materials with full extrusion temperatures

Fused filament fabrication of continuous optic fiber reinforced polylactic acid composites

Effects of extruder temperatures and raster orientations on mechanical properties of the FFF-processed polylactic-acid (PLA) material

Multi-scale damage analysis and fatigue behavior of PLA manufactured by fused deposition modeling (FDM)

Additive manufacturing a powerful tool for the aerospace industry

Author Response

(1) The introduction needs some improvements. Use more recent references for writing this section.

Answer: As suggested, we have updated and included a larger number of recent references in the introduction section.

 (2) There is a lake of information about the following topics: Economical study of the process, Post-processing, Usual defects, Thermal analysis of process

Answer: As suggested, we have included additional paragraphs dealing with economics, post-processing and typical defects in DLP 3D printed parts (chapter 7). We have also included some results from thermal and mechanical analysis of the multi-material vat photopolymerization 3D-printing process (chapter 7).

(3) Please compare this process with other methods such as MEX.

Answer: In the original manuscript, the pros and cons of the individual printing techniques have already been discussed in detail. Thus, in our opinion sufficient information has been provided for the readers.  However, we have introduced a section of a general comparison in chapter 7 and summarized them again in the conclusions.

 (4) Please add some information about the mechanical properties of the sample produced by this method.

Answer: We have added a paragraph discussing the mechanical properties of DLP 3D printed photopolymers (chapter 7).

(5) The conclusion needs more work, please add some information about the advantages and limitations of this process compared with other polymer 3D printing techniques.

Answer: As suggested, we have shortly summarized the advantages and disadvantages of DLP 3D printing compared to other established printing techniques in the conclusion section.

(6) Following papers have been suggested for introduction and comparing results:

Experimental investigations into abrasive flow machining (AFM) of 3D printed ABS and PLA parts

Temperature-compensated constitutive model of fused filament fabrication 3D printed PLA materials with full extrusion temperatures

Fused filament fabrication of continuous optic fiber reinforced polylactic acid composites

Effects of extruder temperatures and raster orientations on mechanical properties of the FFF-processed polylactic-acid (PLA) material

Multi-scale damage analysis and fatigue behavior of PLA manufactured by fused deposition modeling (FDM)

Additive manufacturing a powerful tool for the aerospace industry

Answer: Thank you for suggesting various papers. We have extracted relevant information from the papers and cited all of them in our revised manuscript (chapter 7).

Reviewer 2 Report

In the present review, the author has reviewed the importance of 3D printing of functional materials via vat photopolymerization. The present review is important as there is a lot of progress in the development of functional materials by 3D printing via vat photopolymerization. However, there are few reviews already published recently on this topic. The reviewer thinks that the author has not done an intensive review on this topic and has not covered any most recent reports. I would like not to recommend this review for publication in the present form.

 References:

1.     Zhang, Feng, Liya Zhu, Zongan Li, Shiyan Wang, Jianping Shi, Wenlai Tang, Na Li, and Jiquan Yang. "The recent development of vat photopolymerization: A review." Additive Manufacturing 48 (2021): 102423.

2.     Pagac, M., Hajnys, J., Ma, Q. P., Jancar, L., Jansa, J., Stefek, P., & Mesicek, J. (2021). A review of vat photopolymerization technology: Materials, applications, challenges, and future trends of 3D printing. Polymers13(4), 598.

Author Response

(1) In the present review, the author has reviewed the importance of 3D printing of functional materials via vat photopolymerization. The present review is important as there is a lot of progress in the development of functional materials by 3D printing via vat photopolymerization. However, there are few reviews already published recently on this topic. The reviewer thinks that the author has not done an intensive review on this topic and has not covered any most recent reports. I would like not to recommend this review for publication in the present form.

 References:

  1. Zhang, Feng, Liya Zhu, Zongan Li, Shiyan Wang, Jianping Shi, Wenlai Tang, Na Li, and Jiquan Yang. "The recent development of vat photopolymerization: A review." Additive Manufacturing 48 (2021): 102423.
  2. Pagac, M., Hajnys, J., Ma, Q. P., Jancar, L., Jansa, J., Stefek, P., & Mesicek, J. (2021). A review of vat photopolymerization technology: Materials, applications, challenges, and future trends of 3D printing. Polymers, 13(4), 598.

Answer: As suggested, we have included additional recent literature on “vat photopolymerization 3D printing” in the revised manuscript (incl. the two suggested references).

We also value your opinion about our manuscript, but we would like to draw your attention that our review is strongly focusing on the multi-material aspect and introducing additional functions in multi-material structures (e.g., self-healing), which has not been discussed in that many review publications, yet.

Reviewer 3 Report

Comments on polymers-1752213

The authors provided a comprehensive comparison of additive manufacturing technologies and detailed knowledge of vat photopolymerization in this manuscript. Moreover, the authors extensively reviewed the material chemistry and representative works. This manuscript was written in a well-organized structure and the contents are easy to understand. I recommend the publication of this manuscript after minor revisions.

Major Points:

The authors reviewed several papers reporting hydrogels for biomedical applications. However, in the previous sections, e.g., Section #2 and #4, the authors did not review the manufacturing technologies and chemistries that were commonly used to prepare hydrogels. It would be better if contents related to hydrogels can be added to these sections.

Minor Points:

1. Line 460-461: “… in thicker samples.[128] Fehler! Verweis-quelle konnte nicht gefunden werden.” Please double-check whether this sentence written in Deutsche should be the content for this manuscript.

2. Given the length of this manuscript, it would be better to add a catalog on Page 1.

Author Response

The authors provided a comprehensive comparison of additive manufacturing technologies and detailed knowledge of vat photopolymerization in this manuscript. Moreover, the authors extensively reviewed the material chemistry and representative works. This manuscript was written in a well-organized structure and the contents are easy to understand. I recommend the publication of this manuscript after minor revisions.

Major Points:

(1) The authors reviewed several papers reporting hydrogels for biomedical applications. However, in the previous sections, e.g., Section #2 and #4, the authors did not review the manufacturing technologies and chemistries that were commonly used to prepare hydrogels. It would be better if contents related to hydrogels can be added to these sections.

Answer: As suggested, we included manufacturing technologies for hydrogels in chapter 4.

Minor Points:

(2) 1. Line 460-461: “… in thicker samples.[128] Fehler! Verweis-quelle konnte nicht gefunden werden.” Please double-check whether this sentence written in Deutsche should be the content for this manuscript.

Answer: We have corrected the mistake.

(3) 2. Given the length of this manuscript, it would be better to add a catalog on Page 1.

Answer: We have added a table of content on page 1. 

Round 2

Reviewer 1 Report

Nice work, the paper is ready for the publication process.

Reviewer 2 Report

The revised version can be accepted.